# Altered receptor binding, antibody evasion and retention of T cell recognition by the SARS-CoV-2 XBB.1.5 spike protein

Dhiraj Mannar[1,6], James W. Saville[1,6], Chad Poloni[2,6], Xing Zhu[1], Alison Bezeruk [1], Keith Tidey[1], Sana Ahmed[1], Katharine S. Tuttle[1], Faezeh Vahdatihassani [1], Spencer Cholak[1], Laura Cook [2,3,4], Theodore S. Steiner[2] & Sriram Subramaniam [1,5] ✉

The XBB.1.5 variant of SARS-CoV-2 has rapidly achieved global dominance and exhibits a high growth advantage over previous variants. Preliminary reports suggest that the success of XBB.1.5 stems from mutations within its spike glycoprotein, causing immune evasion and enhanced receptor binding. We present receptor binding studies that demonstrate retention of binding contacts with the human ACE2 receptor and a striking decrease in binding to mouse ACE2 due to the revertant R493Q mutation. Despite extensive evasion of antibody binding, we highlight a region on the XBB.1.5 spike protein receptor binding domain (RBD) that is recognized by serum antibodies from a donor with hybrid immunity, collected prior to the emergence of the XBB.1.5 variant. T cell assays reveal high frequencies of XBB.1.5 spike-specific CD4⁺ and CD8⁺ T cells amongst donors with hybrid immunity, with the CD4⁺ T cells skewed towards a Th1 cell phenotype and having attenuated effector cytokine secretion as compared to ancestral spike protein-specific cells. Thus, while the XBB.1.5 variant has retained efficient human receptor binding and gained antigenic alterations, it remains susceptible to recognition by T cells induced via vaccination and previous infection.

In late 2021, the emergence of the original Omicron SARS-CoV-2 variant (BA.1) ushered in a new chapter of the COVID-19 pandemic. While previously emerged variants (Alpha, Beta, Gamma, Delta) contained up to 10 mutations in their spike glycoproteins, Omicron variants contained an unprecedented >30 spike mutations[1]. Given the role of the spike protein as the major immunogen within SARS-CoV-2 vaccines, the primary consequence of the high number of mutations within the Omicron variants was reduced vaccine efficacy[2–4]. Secondary con-

sequences of Omicron mutations include altered viral tropism and the acquired ability to engage several mammalian ACE2 receptors (mouse, rat, bat, etc.)[5–7]. Specifically, early Omicron lineage spike proteins (BA.1 and BA.2) acquired the ability to bind mouse ACE2 (mACE2) with high affinity, generating the hypothesis of spillover transmission into mice followed by spillback transmission into humans as underlying the emergence of these highly mutated lineages. Throughout 2022, several sub-lineages—BA.1.1, BA.2, and BA.5—successively supplanted the

[1]Department of Biochemistry and Molecular Biology, University of British Columbia, Vancouver, BC V6T 1Z3, Canada. [2]Department of Medicine and BC Children's Hospital Research Institute, University of British Columbia, Vancouver, BC V5Z 4H4, Canada. [3]Department of Microbiology and Immunology, University of Melbourne at the Peter Doherty Institute for Infection and Immunity, Melbourne VIC 3000, Australia. [4]Department of Critical Care, Melbourne Medical School, University of Melbourne, Parkville VIC 3010, Australia. [5]Gandeeva Therapeutics, Inc., Burnaby, BC V5C 6N5, Canada. [6]These authors contributed equally: Dhiraj Mannar, James W. Saville, Chad Poloni. ✉e-mail: sriram.subramaniam@ubc.ca

original BA.1 variant, with a high degree of mutational plasticity observed within the amino terminal domain (NTD) and receptor binding domain (RBD) (Fig. 1A). In late 2022, a recombination occurred between Omicron sub-lineages BA.2.75 and BA.2.10.1 resulting in a new variant XBB.1 and its further sub-lineage XBB.1.5, the latter of which rapidly became the most dominantly sequenced lineage in the United States (Fig. 1B). Given that XBB.1.5 is the first recombinant SARS-CoV-2 lineage to achieve global dominance, and the stark growth advantage it has over earlier Omicron sub-lineages, we investigated the XBB.1.5 spike glycoprotein from the perspectives of receptor binding, antibody evasion, and T cell specificity, and report our findings here. We find that while the XBB.1.5 maintains a high binding affinity for human ACE2 as similar to previous variants, it has a diminished affinity for mouse ACE2, representing a departure from earlier SARS-CoV-2 variants. We show that despite significant antibody escape, the XBB.1.5 spike protein is still recognized by pre-existing antibodies and T cells from donors with hybrid immunity.

## Results

### Unaltered architecture of the XBB.1.5 spike protein

We first characterised the overall architecture of the XBB.1.5 spike protein via cryogenic-electron microscopy (cryo-EM). We obtained a global reconstruction of the XBB.1.5 spike ectodomain at 2.8 Å resolution, finding 2 down RBDs and 1 unresolved RBD (Fig. 1C, D). The architecture of the XBB.1.5 spike protein is similar to previous variants such as the Alpha, Beta, Gamma, Delta, and Omicron lineages[5,8–13]. The resolution of the RBD and NTD regions of the XBB.1.5 spike protein is much lower than that of the S2 core (Supplementary Fig. 1), highlighting the flexibility of these regions, as seen with previous variants. Tamura et al.[14] recently reported the cryo-EM structure of the closely related XBB.1 spike protein, resolving two different closed states (all RBDs down)[14]. The major difference between these two closed states is slight differential positioning of the RBDs in the down state, suggesting RBD flexibility as observed in the unresolved RBD within the XBB.1.5 spike structure presented in Fig. 1C, D.

### Enhanced binding of human ACE2 by the XBB.1.5 spike protein

Recent reports have demonstrated enhanced receptor binding by the XBB.1.5 spike protein compared to its XBB.1 predecessor[15–17]. We performed biolayer interferometry (BLI) experiments, finding an overall trend consistent with increased human ACE2 (hACE2) affinity for the XBB.1.5 RBD compared with XBB.1, although in our experimental setup this difference did not reach the threshold for statistical significance (Supplementary Fig. 2A). Nevertheless, it has been demonstrated that the reported increase in affinity is due to the mutation of residue 486 to proline in XBB.1.5 instead of serine in XBB.1[15–17]. Importantly, previous variants—including BA.2—harbour a phenylalanine at this position which makes direct contacts with hACE2. We compared the hACE2 binding affinity of the XBB.1.5 RBD as measured via BLI, to the wild-type (Wuhan-Hu-1 with the D614G mutation, WT) and BA.2 RBDs (Fig. 2A). Consistent with previous reports, we observed potent (nanomolar) binding of the XBB.1.5 RBD to hACE2. When compared to WT, the affinity of the XBB.1.5 spike for hACE2 is enhanced to a similar extent as BA.2, which we and others have previously characterised[5,6,18] (Fig. 2A). To further complement our BLI analysis we utilized surface plasmon resonance (SPR) to measure the hACE2 affinity of WT, BA.2, and XBB.1.5 RBDs, finding similar enhancements in binding potency for the BA.2 and XBB.1.5 RBDs (Fig. 2B).

To explore the structural basis of this binding enhancement, we performed cryo-EM studies of the complex formed between XBB.1.5 and hACE2. We observed a 3D class of this complex with full occupancy of hACE2 on the trimer, achieving a global reconstruction of this complex at 2.5 Å resolution using C3 symmetry (Fig. 2C). This is in contrast with our studies of previous SARS-CoV-2 variants, finding only one or two RBDs bound with varying density strengths corresponding to hACE2 (with samples prepared at identical spike:hACE2 molar ratios), and is consistent with the premise of enhanced hACE2 binding by the XBB.1.5 spike protein. Local refinement enables visualisation of the XBB.1.5 RBD - hACE2 interface at 2.8 Å resolution, permitting unambiguous placement of P486 (Fig. 2D, E). Inspection of the P486 mutation reveals that it does not alter the conformation of the loop

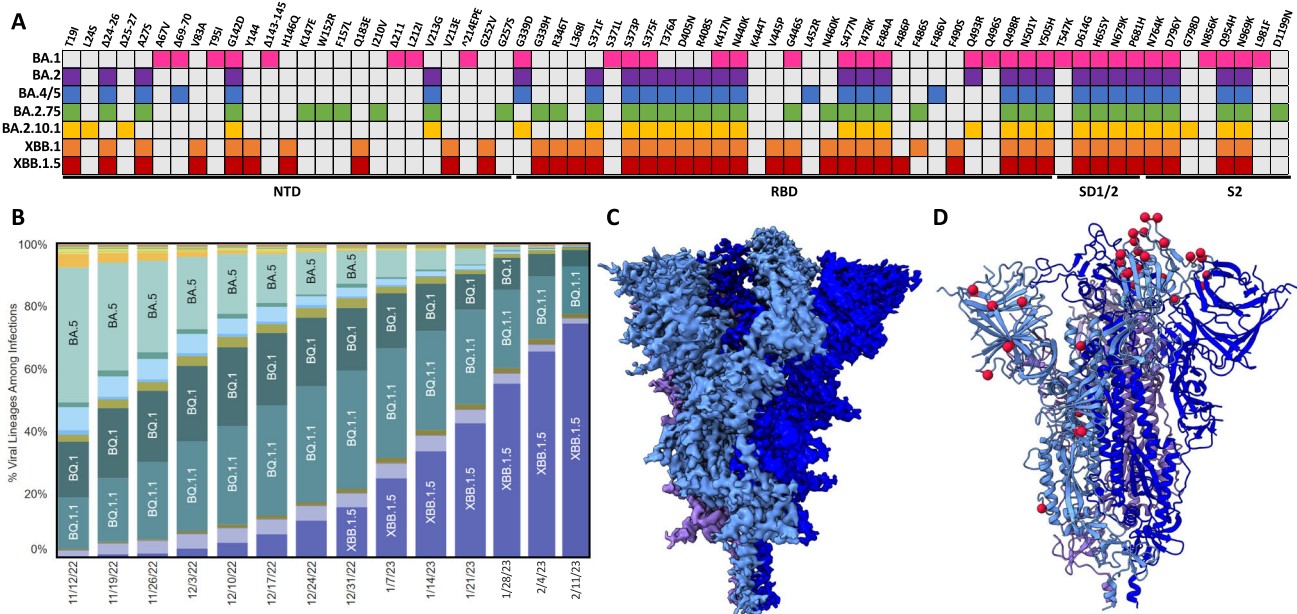

**Fig. 1 | Mutational profile and prevalence of the XBB.1.5 SARS-CoV-2 lineage.**
**A** Amino acid mutations in the spike glycoprotein open reading frame for various Omicron and XBB sub-lineages. Each coloured box represents a mutation within a specific omicron sub-lineage. NTD amino terminal domain, RBD receptor binding domain. **B** Weekly weighted estimates of lineage proportion from sequencing data in the United States[39]. Weeks from 28 Jan–11 Feb 2023 represent Nowcast estimates which consistently align with the weighted proportions based on reported sequencing data. **C** CryoEM density map of the XBB.1.5 spike protein, with each protomer coloured a shade of blue or purple. **D** Resultant atomic model of the XBB.1.5 spike protein with modelled mutational locations denoted with red spheres on a single protomer.

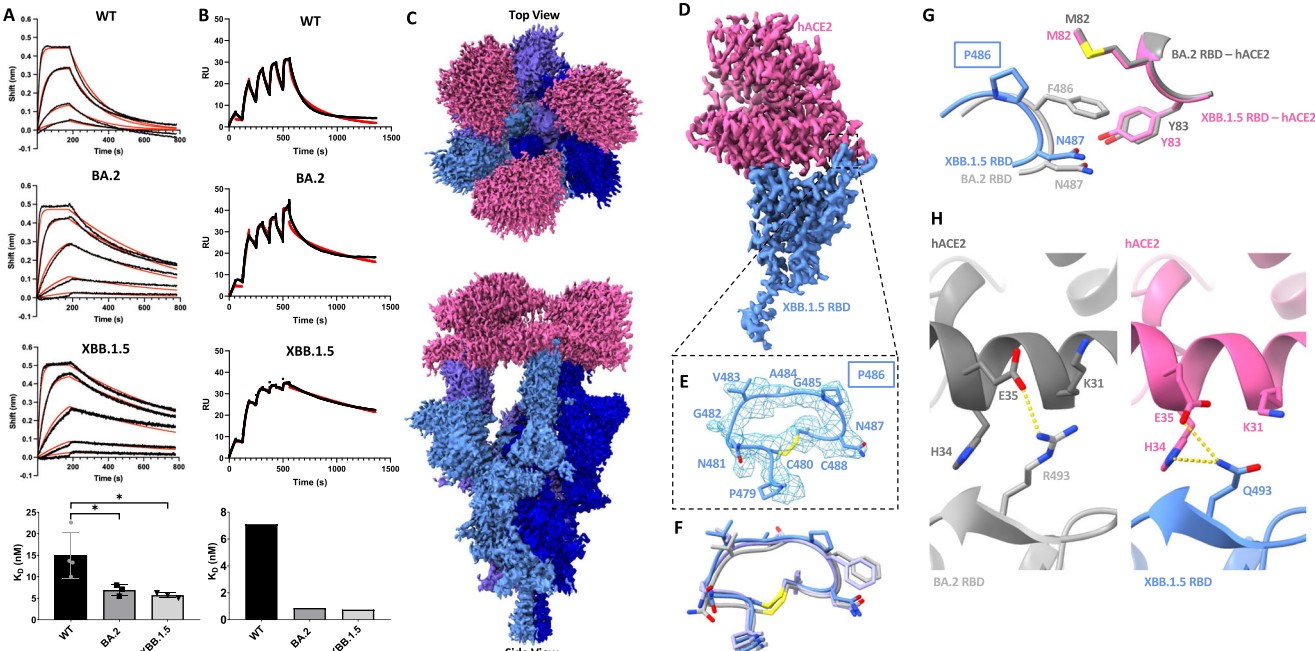

**Fig. 2 | Analysis of human ACE2 (hACE2) engagement by the XBB.1.5 spike protein. A** Biolayer interferometry analysis of WT, BA.2, and XBB.1.5 RBDs binding to immobilized dimeric hACE2. Black curves represent raw data which were fit to a model using a 1:1 binding stoichiometry (red) to determine the reported dissociation constants. Experiments were performed 4 times ($n = 4$) for the WT and 3 times ($n = 3$) for the remaining variants, and a representative curve is shown for each condition. Results are summarized at the bottom; error bars denote the standard deviation. Statistical significance was assessed via ANOVA with Dunnett's post test for multiple comparisons against the WT (*$P \leq 0.05$), WT vs BA.2 ($P = 0.0412$), WT vs XBB.1.5 ($P = 0.0228$). **B** Single-cycle kinetic analyses of WT, BA.2, and XBB.1.5 RBDs binding to immobilized dimeric hACE2 measured via surface plasmon resonance. Black curves represent raw data which were fit to a model using a 1:1 binding stoichiometry (red) to determine the reported dissociation constants which are tabulated below from a single experiment. RU: Response units. Experiments were performed one time. **C** Global Cryo-EM density map of the XBB.1.5 spike−hACE2 complex. **D** Local Cryo-EM density map of the hACE2−XBB.1.5 RBD region. **E** Map and model of the hACE2 binding ridge loop in the XBB.1.5 RBD when bound to hACE2. **F** Comparison of the hACE2 binding ridge loop region between XBB.1.5, BA.2, and WT RBDs when bound to ACE2. **G** Comparison of the hACE2 contacts made by RBD residues 486 and 487 in the XBB.1.5 and BA.2 RBD. **H** Comparison of the hACE2 contacts made by residue 493 within BA.2 and XBB.1.5 RBDs. Dashed lines indicate potential interactions (salt bridges or hydrogen bonds). Models were aligned by the RBD for all superpositions. PDB ID: 6M0J and 8DM6 were used for the WT-ACE2 complex and BA.2-ACE2 complex, respectively.

region within the hACE2 binding ridge[19], compared to BA.2 or WT RBDs when bound to ACE2 (Fig. 2F). Additionally, P486 does not restore the hydrophobic hACE2 contacts made by F486 (Fig. 2G). We note that while the density for the M82 sidechain is strong within the BA.2-hACE2 complex (consistent with ordering due to interactions with F486), there is weak density for M82 in the XBB.1.5-hACE2 complex, suggesting that the side chain is flexible and not participating in an interaction. In the absence of any additional hACE2 contacts afforded by P486, and the limited angular degrees of freedom intrinsic to proline within a polypeptide chain, it is conceivable that the described increased hACE2 binding affinity over XBB.1[15,16] is due to conformational stabilisation of the hACE2 binding ridge loop region in an hACE2 binding competent state.

When compared with the hACE2-BA.2 structure, alterations in local contacts made by XBB.1.5 RBD residue 493 are apparent (Fig. 2H). R493 in BA.2 establishes a salt bridge with hACE2 residue E35 and displaces hACE2 residue K31 away from the RBD interface. Q493 within the XBB.1.5 RBD sits within hydrogen bonding distance of hACE2 residue H34 and accommodates space for hACE2 residues E35 and K31 to extend closer towards the RBD interface, with E35 positioned in hydrogen bonding distance of Q493. These alternative hACE2 interactions made by Q493, along with the potential stabilising effect of P486, may contribute favourably to the hACE2 binding energy to compensate for the loss of the R493-E35 salt bridge and the hydrophobic interactions made by F486 with hACE2 residues M82 and Y83. These compensatory gains and losses of intermolecular interactions rationalize the overall comparable hACE2 binding potencies for BA.2 and XBB.1.5 as measured by BLI and SPR. Finally, when compared to

recently published chimeric structures of XBB.1 and XBB.1.5 RBD−hACE2 solved by X-ray crystallography, our cryoEM structure reported here, and the specific positioning of the residues discussed above are highly aligned (Supplementary Fig. 4)[17].

**Attenuation of mouse ACE2 binding by the XBB.1.5 spike protein**
There have been several hypotheses surrounding the origins of the Omicron lineage variants since the emergence and global spread of BA.1 − the first sub-lineage of this group of variants. One prevalent theory involves a zoonotic transmission of BA.1 from a mouse reservoir back into humans[20]. Consistent with this theory, BA.1 and BA.2 Omicron spike proteins exhibited an enhanced binding affinity for mouse ACE2 (mACE2) as compared to any previously emerged SARS-CoV-2 variant[6,21]. We recently reported the cryo-EM structure of the BA.1 and BA.2 Omicron spike proteins in complex with mACE2, finding Omicron mutations Q493R, N501Y, and Y505H to engage non-conserved ACE2 residues between hACE2 and mACE2, likely rationalizing the enhanced binding of mACE2 by these variant spike proteins[5]. While the XBB.1.5 variant contains the N501Y and Y505H mutations, it does not harbour the Q493R mutation, and as such, we hypothesized that the absence of this mutation may impact its ability to engage mACE2 and could represent a departure from the omicron lineages with regards to mACE2 binding. To test this notion, we performed an enzyme-linked immunosorbent assay (ELISA) to detect differences in mACE2 binding between WT, BA.2, and XBB.1.5 spike proteins (Fig. 3A). As expected, this assay recapitulated the previously reported enhanced mACE2 binding by BA.2 spike as compared to WT, while demonstrating substantially decreased - though not abolished - mACE2 binding by the

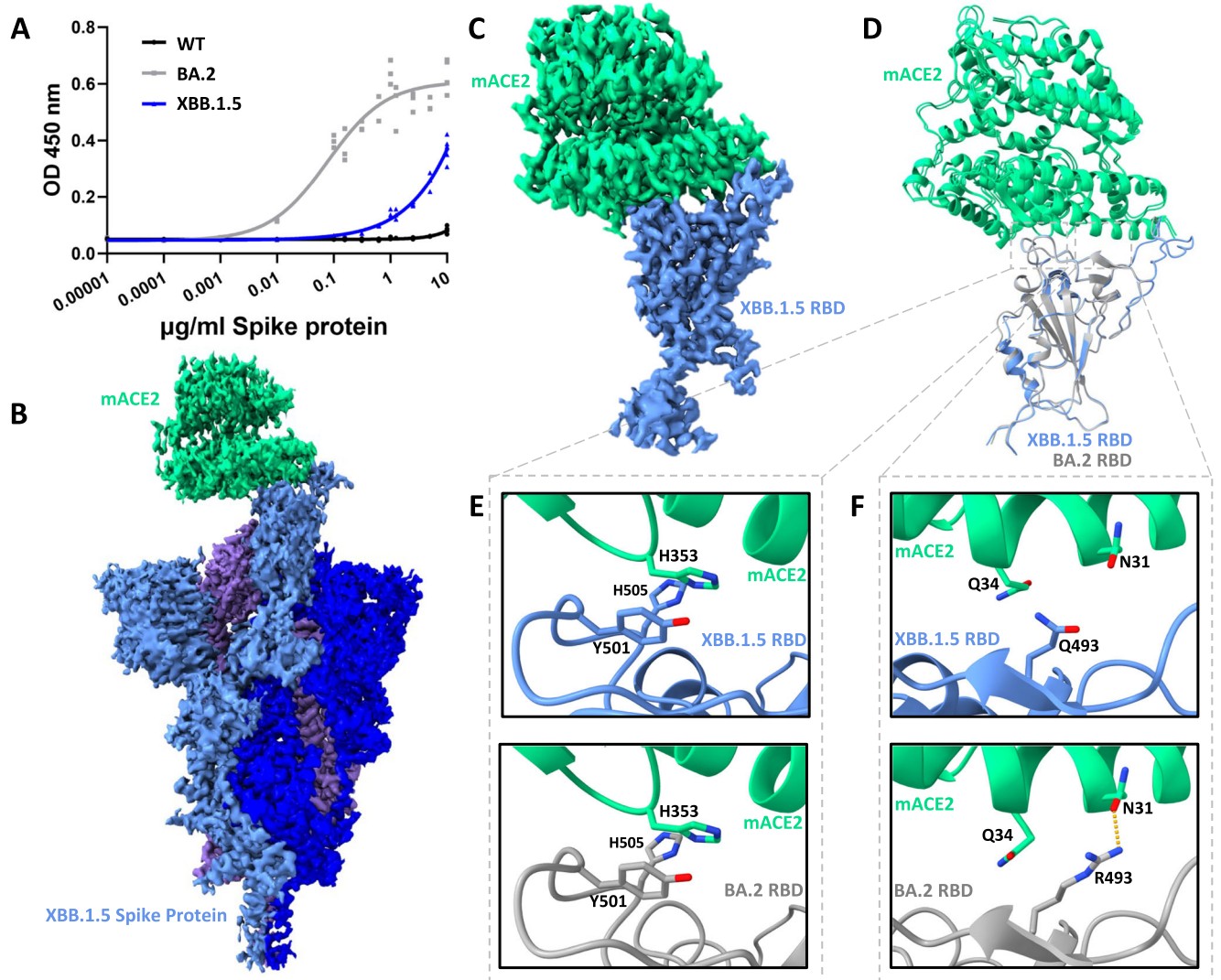

**Fig. 3 | The XBB.1.5 spike protein−mouse ACE2 interaction. A** ELISA of WT, BA.2, and XBB.1.5 spike proteins binding to mACE2. The results of two independent experiments are plotted, with technical triplicates performed in each experiment. OD450 nm: Optical density at 450 nm. **B** CryoEM density of XBB.1.5 spike protein bound to mACE2. The spike protein is shown in shades of blue and purple while mACE2 is shown in green. **C** As in (**B**), but for the focused-refined mACE2−XBB.1.5 RBD complex. **D** The resultant atomic model of the mACE2−XBB.1.5 RBD complex structurally aligned with the mACE2−BA.2 RBD complex (PDB ID: 8DM8). **E** Focused view of mutated residues Y501 and H505 in the mACE2−XBB.1.5 RBD complex (top) and mACE2−BA.2 RBD complex (bottom). **F** Focused view of RBD residue 493 in the mACE2−XBB.1.5 RBD complex (top) and mACE2−BA.2 RBD complex (bottom).

XBB.1.5 spike. This result is consistent with the idea that the Q493R spike protein mutation contributes to enhanced mACE2 binding affinity.

Despite the measured decrease in XBB.1.5 spike−mACE2 binding affinity (Fig. 3A), we were able to solve the cryo-EM structure of the XBB.1.5 spike−mACE2 complex, with the aim to provide a structural rationale for the decreased affinity. Three-dimensional reconstructions, with both one and two mACE2 molecules bound to the XBB.1.5 spike protein, were resolved and our analysis continues here with the slightly higher resolution one mACE2 bound state−with direct parallels possible for the two mACE2 bound state (Fig. 3B). Focused-refinement of the XBB.1.5 RBD bound to mACE2 was possible, allowing for detailed analysis of interactions at this binding interface (Fig. 3C). Figure 3D shows that the positioning of mACE2, relative to the RBD, is preserved between BA.2 and XBB.1.5 RBDs, suggesting that large structural changes do not rationalize the binding affinity differences, rather a dissection of interactions at the binding interface may provide mechanistic insights. Indeed, differential side-chain interactions are observed at amino acid position 493 between the BA.2 and XBB.1.5

RBDs (Fig. 3F). The side-chain of XBB.1.5 residue Q493 is no longer positioned to interact with residue N31 of mACE2, as was observed in the BA.2 RBD−mACE complex. While Q493 may have been expected to gain an interaction with mACE2 residue Q34, this residue is found in a rotamer facing away from Q493 and therefore beyond typical interaction distance (>4 Å). mACE2−RBD interactions at the N501Y and Y505H mutated sites are near identical between BA.2 and XBB.1.5, suggesting preserved affinity at this site (Fig. 3E). In sum, our functional and structural data suggest that the loss of interactions caused by the R493Q revertant mutation lessens XBB.1.5 spike−mACE2 binding affinity, yet the binding affinity is not abolished due to preserved favourable contributions from the conserved N501Y and Y505H mutations.

## Antibody escape by the XBB.1.5 spike protein
Extensive evasion of neutralising antibody activity has been described for the XBB.1.5 variant[15,16], suggesting a substantial shift in the antigenicity of the XBB.1.5 spike protein. We first performed binding analyses using a small panel of RBD- and NTD-directed antibodies,

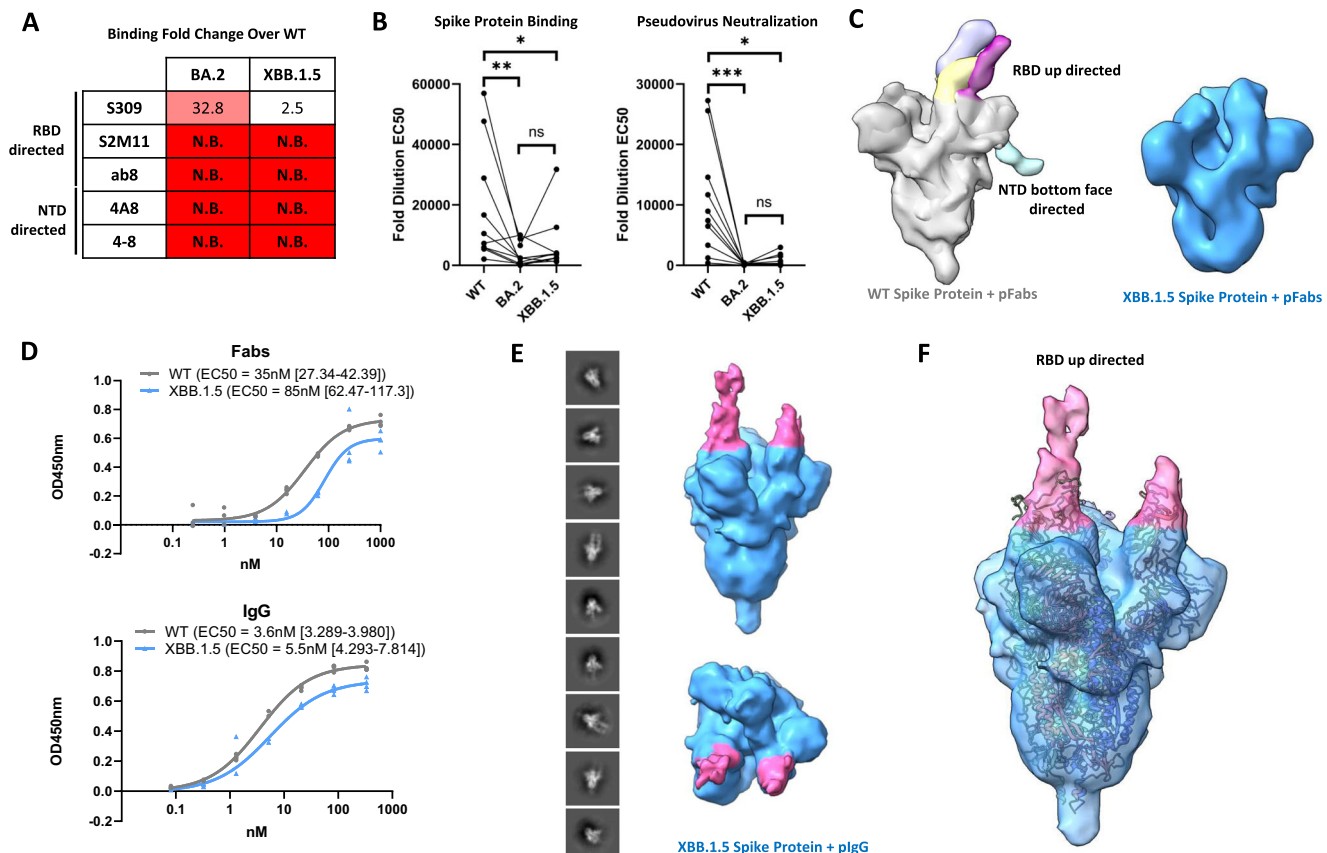

**Fig. 4 | Analysis of antibody binding to the XBB.1.5 spike protein. A** Analysis of monoclonal antibody binding to the WT, BA.2, and XBB.1.5 spike proteins via ELISA. $EC_{50}$ ratios over the WT spike protein binding $EC_{50}$ are shown for each antibody assayed (N.B. no binding). **B** Analysis of serum IgG binding to the WT, BA.2, and XBB.1.5 spike proteins via ELISA (left) and serum neutralization of pseudoviruses harbouring the WT, BA.2, and XBB.1.5 spike proteins (right) from $n = 10$ vaccinated adults. Fold dilution $EC_{50}$ values are plotted for each spike protein. A pairwise statistical significance test was performed using the Friedmans and Dunn's post-test for multiple comparisons (*$P \le 0.05$, **$P \le 0.01$, ***$P \le 0.0001$, ns: not significant). Calculated $P$ values are as follows – spike protein binding: WT vs BA.2 ($P = 0.001$), WT vs XBB.1.5 ($P = 0.0417$), BA.2 vs XBB.1.5 ($P = 0.7907$), pseudovirus neutralization: WT vs BA.2 ($P = 0.0001$), WT vs XBB.1.5 ($P = 0.0304$), BA.2 vs XBB.1.5 ($P = 0.3526$). **C** Negative stain electron microscopy studies of WT and XBB.1.5 spike proteins using polyclonal Fab fragments (pFabs). Top: Superposition of all Fab-

spike protein reconstructions obtained using polyclonal Fabs and the WT spike protein. Maps were superposed in chimeraX and Fab densities are colourized and annotated by general location of the epitope recognized. Bottom: Representative 3D reconstruction of the XBB.1.5 spike trimer showing no additional density corresponding to Fab fragments. **D** ELISA analysis of polyclonal Fabs and IgGs binding to WT and XBB.1.5 spike proteins. Experiments were done in technical quadruplicate ($n = 4$) and are shown as points. EC50 values along with 95% confidence intervals are shown. **E** Negative stain electron microscopy studies of polyclonal IgG (pIgG) binding the XBB.1.5 spike protein. 2D class averages along with both side and top views of the resulting 3D reconstruction are shown. Additional density corresponding to Fab regions are coloured in pink. **F** Fit of a SARS-CoV-2 Spike protein model with all RBDs in the up position (PDB 7X7N with ligands removed) into the obtained 3D reconstruction of an IgG-XBB.1.5 spike protein complex.

measuring binding to the ancestral (WT), BA.2, and XBB.1.5 spike proteins (Fig. 4A and Supplemental Fig. 6A). Like BA.2, the XBB.1.5 spike demonstrated significant evasion of all antibodies with the exception of S309, a precursor to the clinical monoclonal antibody sotrovimab. Our measurements demonstrate retention of S309 binding to the XBB.1.5 spike protein—as compared to BA.2—potentially due to the differential G339D (BA.2) versus G339H (XBB.1.5) mutation, which occurs within the S309 epitope. We further confirmed the retention of S309 binding to the XBB.1.5 variant through an ELISA using minimal RBD constructs (Supplemental Fig. 6B). This finding is consistent with a recent study measuring the neutralization of S309 against SARS-CoV-2 variant pseudoviruses, finding unaltered neutralization activity against BA.2.75 (G339H containing) as compared with BA.2 and BA.4 (both G339D containing)[22]. The authors of this study noted that the change between D339 (BA.2) and H339 (BA.2.75 and XBB.1.5) is a charge-reversing mutation within the S309 epitope making the enhanced neutralization activity relative to D339 containing variants particularly striking. The exact role of amino acid identity at position 339 in S309 escape is further complicated by the R346T and

L368I mutations present within the XBB.1.5 variant, which have previously been reported to drive S309 escape[23].

We next measured polyclonal IgG binding of WT, BA.2, and XBB.1.5 spike proteins using sera from healthy volunteers with varying vaccination and infection histories (Supplementary Table 2). We found both BA.2 and XBB.1.5 spike proteins to exhibit considerable decreases in IgG binding from these samples, with a greater spread in XBB.1.5 binding potencies (Fig. 4B and Supplementary Fig. 6C). We additionally measured serum neutralization of pseudoviruses harboring WT, BA.2, and XBB.1.5 spike proteins using these sera samples and found both BA.2 and XBB.1.5 pseudoviruses to be less sensitive to serum neutralization when compared to WT (Fig. 4B). Thus, both spike protein binding and pseudoviral neutralization assays provide data consistent with the strong antibody evasion reported for both of these variants[15,16].

Despite the significant loss of serum IgG potency, all serum samples exhibited binding of the XBB.1.5 spike protein, suggesting the presence of conserved epitopes within the XBB.1.5 spike protein which may be targeted by pre-existing serum antibodies. To gain a structural

understanding of the antigenicity of the XBB.1.5 spike protein and to identify any of these preserved epitopes, we proceeded to employ negative stain electron microscopy, which has been utilised previously to map out polyclonal Fab fragments on several viral spike proteins[24–27]. We selected serum from a quadruple-vaccinated 24-year-old male donor (ID 4 in Supplementary Table 2) who received three monovalent doses followed by one BA.1 bivalent dose and had a history of COVID-19. We first performed an electron microscopy analysis on immune complexes generated using the ancestral WT spike protein, finding robust epitope coverage across the immunodominant RBD and NTD regions (Fig. 4C and Supplementary Fig. 7A). We identified several unique classes of Fab fragments which target the RBD in the up conformation at distinct sites, along with a NTD "bottom face" targeting Fab fragment, demonstrating the structural coverage achieved by vaccine- and infection-elicited antibodies. When performing an identical experiment using the XBB.1.5 spike protein, we failed to recover any Fab fragment densities (Fig. 4C and Supplementary Fig. 7B), highlighting the antigenic alterations within these epitopes, and consistent with the antibody evasive nature of the XBB.1.5 variant.

We next utilised intact IgG molecules for electron microscopy studies hypothesising that the avidity effect might help overcome antibody evasive mutations and permit stronger binding of the XBB.1.5 spike protein. Consistent with this hypothesis, measurements of polyclonal Fab fragment binding demonstrated a larger drop in potency between WT and XBB.1.5 spike proteins as compared to intact polyclonal IgGs from the same sample (Fig. 4D). We observed varying extents of IgG binding to the XBB.1.5 spike protein as evidenced by the presence of multiple species when a mixture was subjected to size exclusion chromatography (Supplementary Fig. 7C). We collected data from species exhibiting mild and no immune crosslinking and generated an IgG-XBB.1.5 spike protein reconstruction from the pooled data set. The resulting reconstruction shows additional densities above the RBD regions of the trimer, with one density being much stronger than the other (Fig. 4E). This is consistent with an RBD-directed IgG antibody bound to the trimer, with perhaps some alignment error distributing the Fab densities unevenly across each RBD. Nevertheless, this reconstruction permits assignment of the RBD in the up position as the target of this IgG (Fig. 4F). When the model for the hACE2-XBB.1.5 spike protein complex is fit into this reconstruction, it is apparent that this IgG binds the RBD in an ACE2 competitive manner, a common mechanism of SARS-CoV-2 neutralising antibodies[28] (Supplementary Fig. 7D). Thus, this finding highlights the ability for pre-existing serum IgG molecules to recognize the XBB.1.5 RBD at potentially neutralising epitopes.

### Alteration of T cell reactivity towards the XBB.1.5 spike protein in vaccinees

Given the antibody evasive nature of the XBB.1.5 spike protein, we sought to probe the cellular arm of the adaptive immune system for recognition of the XBB 1.5 spike protein. To this end, we obtained peripheral blood mononuclear cells (PBMCs) from the donors outlined in Supplementary Table 2, and conducted stimulation studies using the WT, BA.2, and XBB.1.5 spike proteins (Fig. 5). After a 44-hour stimulation, antigen-specific CD4+ T cells were detected by induced co-expression of CD25 and OX40 (CD134) and CD8+ T cells by CD137 (4-1BB) and CD69 co-expression (gating strategy in Supplementary Fig. 8)[29]. Ancestral spike-specific CD4+ T cell responses were detected in 90% of individuals, while 80% and 60% of individuals had detectable BA.2 and XBB.1.5 spike-specific CD4+ T cell responses respectively (Fig. 5A). CD8+ T cell responses were detected in 90% of individuals and did not differ between spike variants (Fig. 5B). Further, there were no statistically significant differences in the overall magnitude of T cell responses to the spike variants, being an average of 0.73% (SD = 2.48) of circulating CD4+ T cells and 2.03% (SD = 3.72) of CD8 T+ cells (Fig. 5A,

B). Next, spike-specific CD4+ T cell responses were phenotypically characterised to quantify the relative proportions of T helper (Th) cell subsets. Th subsets were identified based on their surface expression of CCR4, CCR6, CXCR3, CXCR5, and PD-1 (gating strategy in Supplementary Fig. 9). XBB.1.5 spike-specific CD4+ T cells contained significantly more Th1 cells as compared to ancestral and BA.2 spike responses (Fig. 5C). Further, responses to ancestral spike had decreased proportions of CCR6+ T cells and appeared to skew towards a Th2 phenotype. BA.2 spike-specific CD4+ T cells predominantly consisted of cells with a Th9-like phenotype (CXCR3+CCR4negCCR6+), with proportions of these cells being significantly higher than within ancestral or XBB.1.5 spike-specific responses (Fig. 5C). There were no significant differences in circulating T follicular helper (cTfh) subsets between responses to the spike variants (Supplementary Fig. 10). Thus, our data demonstrate recognition of the XBB.1.5 spike protein by pre-existing memory CD8+ and CD4+ T cells, with increased variability and a skewing towards a Th1 phenotype in XBB.1.5 spike-specific CD4+ T cells across donors.

We also collected assay supernatants from these PBMC cultures after 44-hours of stimulation with spike protein, and measured the concentrations of IFNγ, TNF, IL-13, IL-17A, IL17F, IL-22, IL-2, IL-10, IL-4, IL-5 and IL-9. Overall, supernatants from XBB.1.5-stimulated PBMCs had lower concentrations of each cytokine detected, with significantly lower concentrations of IFNγ and IL-22 as compared to both ancestral and BA.2 spike stimulated PBMCs (Fig. 5D). Both XBB.1.5 and BA.2 spike stimulated cells secreted less TNF than ancestral spike stimulated cells. XBB.1.5 spike stimulated cells had significantly lower levels of IL-13 and IL-17F compared to ancestral spike stimulated cells, with no statistically significant differences seen for IL-2, Il-4, IL-5, IL-9 and IL-10 (Supplementary Fig. 11). Taken together, these data demonstrate that circulating, pre-existing spike-specific memory CD8+ and CD4+ T cells in donors with hybrid immunity are capable of recognizing the XBB.1.5 spike protein, and that these cells mount measurable, albeit somewhat attenuated, cytokine secretion in response to spike stimulation.

## Discussion

The XBB.1.5 variant is supplanting previous SARS-CoV-2 lineages worldwide. Herein, we have presented structural and biochemical analyses of the XBB.1.5 spike protein with a focus on human and mouse ACE2 binding, monoclonal and polyclonal antibody evasion, and T cell responses. Our major findings include: (1) increased XBB.1.5 spike - hACE2 affinity as compared to WT spike and similar affinity as compared to other Omicron variant spikes; (2) decreased, but not abolished XBB.1.5 spike - mACE2 affinity, likely as a result of the R493Q revertant mutation; (3) significant escape from monoclonal and polyclonal antibodies, consistent with previously emerged Omicron variants; and (4) preserved spike-specific T cell frequencies as compared to both the WT and BA.2 spike proteins, but with a qualitative skew towards Th1 cells, and a reduction in cytokine production following antigen stimulation. The synthesis of these findings supports a viral fitness advantage afforded by the XBB.1.5 spike protein and rationalizes, in part, its recent growth advantage over other SARS-CoV-2 lineages.

As mentioned previously, the original Omicron sub-lineages (BA.1 and BA.2) acquired the ability to engage the mouse ACE2 receptor. Additionally, mouse-adapted SARS-CoV-2, generated by serial passage of WT virus in mice, selected for mutations at positions Q493 and Q498, which are both mutated sites in BA.1 and BA.2 spike proteins[30,31]. Successive Omicron sub-lineages (BA.4 and BA.5) and recombination variants (XBB.1 and XBB.1.5) have since lost the Q493R mutation (Fig. 1A), which we demonstrate in Fig. 3 to result in decreased mACE2 affinity. The Q493R mutation was perhaps important to enable the SARS-CoV-2 virus to spillover into mice (or other mammals), sampling different selection pressures in its new host(s), resulting in the hyper-

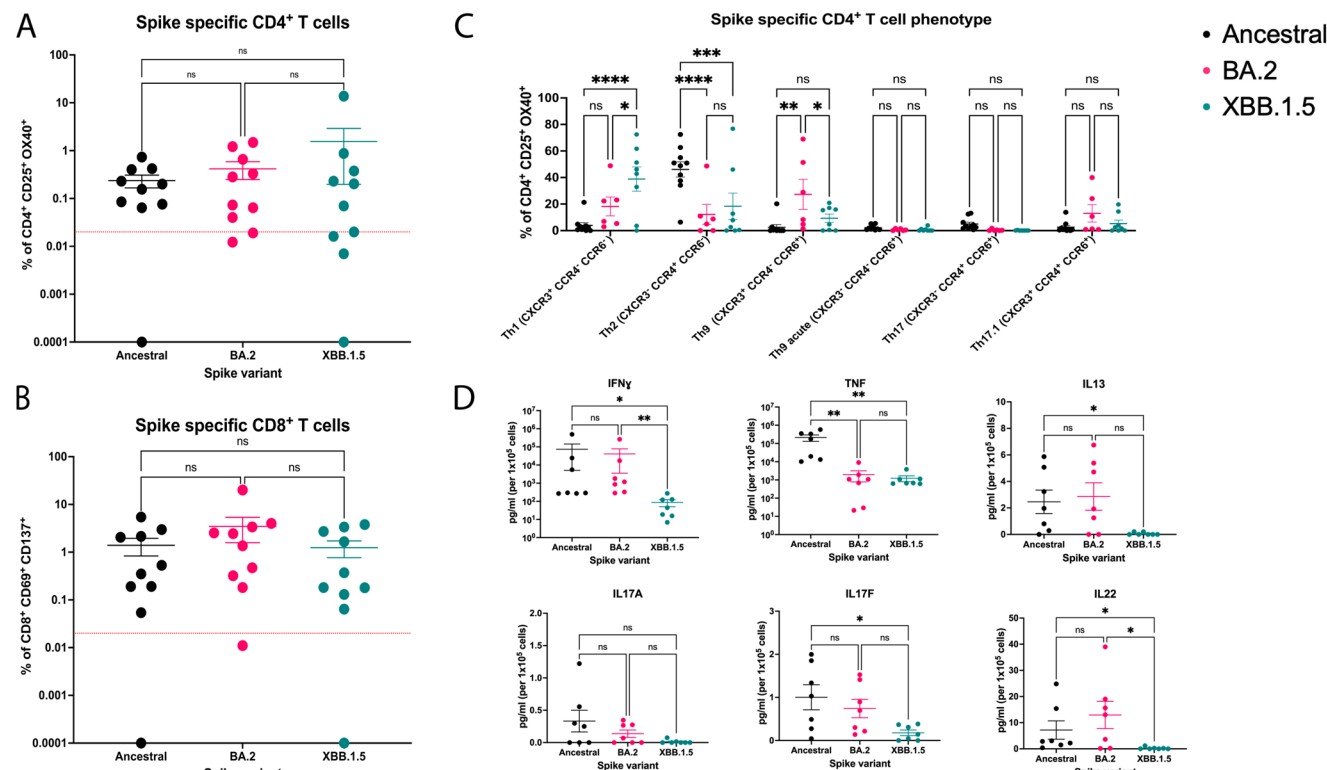

**Fig. 5 | T cell responses to ancestral (WT), BA.2 and XBB.1.5 SARS-CoV-2 spike proteins.** After 44-hours of stimulation with antigen we quantified the **A** frequency of spike protein specific CD4+ T cells (CD25+OX40+) and **B** frequency of spike protein specific CD8+ T cells (CD137+CD69+) in PBMCs from n = 10 vaccinated adults for three spike variants. Kruskal–Wallis tests with Dunn's multiple comparisons tests were used to determine significant differences in T cell responses (ns not significant). **C** Phenotypic analysis of spike specific CD4+ T cells from (**A**), responses were of sufficient magnitude to quantify subsets for n = 10 ancestral, n = 6 BA.2 and n = 8 XBB.1.5 spike stimulated assays. A two-way ANOVA with Tukey's multiple comparisons test was used to determine significant differences in CD4+ T cell phenotypes between spike variants. (*P ≤ 0.05, **P ≤ 0.01, ***P ≤ 0.0001, ns: not significant). Calculated P values are as follows: Th1: WT vs BA.2 (P = 0.1111), WT vs XBB.1.5 (P < 0.0001), BA.2 vs XBB.1.5 (P = 0.0164). Th2: WT vs BA.2 (P < 0.0001), WT vs XBB.1.5 (P = 0.0001), BA.2 vs XBB.1.5 (P = 0.6877). Th9: WT vs BA.2 (P = 0.0018),

WT vs XBB.1.5 (P = 0.5457), BA.2 vs XBB.1.5 (P = 0.0427). Th9 acute: WT vs BA.2 (P = 0.9742), WT vs XBB.1.5 (P = 0.9732), BA.2 vs XBB.1.5 (P > 0.9999). Th17: WT vs BA.2 (P = 0.7978), WT vs XBB.1.5 (P = 0.7335), BA.2 vs XBB.1.5 (P = 0.9988). Th17.1: WT vs BA.2 (P = 0.3035), WT vs XBB.1.5 (P = 0.9065), BA.2 vs XBB.1.5 (P = 0.5498). **D** Cytokine concentrations in supernatants collected from spike-stimulated wells at 44 hours were quantified as pg/mL per $10^5$ cells for n = 7 individuals. A two-way ANOVA with Tukey's multiple comparisons test was used to determine significant differences in cytokine levels. (*P ≤ 0.05, **P ≤ 0.01, ns not significant). Calculated P values are as follows: IFNγ: WT vs BA.2 (P > 0.999), WT vs XBB.1.5 (P = 0.0153), BA.2 vs XBB.1.5 (P = 0.0032). TNF: WT vs BA.2 (P = 0.0050), WT vs XBB.1.5 (P < 0.0043), BA.2 vs XBB.1.5 (P > 0.9999). IL-13: WT vs BA.2 (P > 0.9999), WT vs XBB.1.5 (P = 0.0445), BA.2 vs XBB.1.5 (P = 0.0676). IL-17A: P = 0.2188). (IL-17F: WT vs BA.2 (P = 0.6622), WT vs XBB.1.5 (P < 0.0330), BA.2 vs XBB.1.5 (P = 0.1723). IL-22: WT vs BA.2 (P > 0.9999), WT vs XBB.1.5 (P = 0.0221), BA.2 vs XBB.1.5 (P = 0.0114).

mutated original Omicron variant[5,32,33]. Since spilling back into humans, the Omicron variant may no longer be subjected to the selective pressure to bind mACE2, thus resulting in the revertant R493Q mutation. Our data, when synthesized with the broader literature, suggests that increased mACE2 binding affinity may have been important for originating the Omicron variant, but since its spill back into humans, residual mACE2 binding is only a relic of its zoonotic past.

As with most structural biology approaches that characterize antibody–antigen interactions, electron microscopy typically requires the use of Fab fragments to avoid the formation of structurally intractable higher order immune complexes—caused by the avidity effects of bivalent IgG molecules. However, here we have utilised intact IgG molecules in our electron microscopy studies, given our inability to visualise Fab fragments bound to the XBB.1.5 spike protein—which was likely a result of mutational escape (Fig. 4). We were unable to recover any of the WT RBD or NTD directed Fab fragments bound to the XBB.1.5 spike protein using bulk Fabs, but we could obtain a reconstruction of the XBB.1.5 trimer with additional Fab densities when using bulk IgGs, highlighting the phenomenon of avidity-mediated binding of variant spike proteins within human sera. The RBD footprint bound in this reconstruction overlaps with the ACE2

binding region, highlighting a neutralizing role for pre-existing serum IgG molecules that are capable of cross-reacting with the XBB.1.5 spike protein. Taken together, these results demonstrate the ability of polyclonal antibodies to exploit avidity to overcome mutational effects within SARS-CoV-2 variant of concern spike proteins. We note that while the XBB.1.5 spike protein demonstrates evasion of antibody binding and neutralization, potential effector mechanisms mediated by serum antibodies have not been thoroughly assessed in the present study and may be of importance for viral clearance.

Here we have provided a comparison of the frequency and phenotype of ancestral, BA.2, and XBB.1.5 spike-specific T cells in healthy, vaccinated individuals (Fig. 5). In this report we analyzed TNF, IL-13, IL-17F, and IL-22 production following XBB.1.5 spike stimulation, all of which we found were at decreased concentrations compared to ancestral spike stimulation. Notably, IL-22 and IL-22-producing CD4+ T cells are robustly induced by SARS-CoV-2 infection and Omicron spike peptides, are implicated in tissue repair, and are hypothesized to contribute to lung healing following infection[34–36]. Decreased levels of IL-22 in response to XBB.1.5 spike stimulation may have implications in lung healing. However, it is not known if these decreases are clinically relevant, as all samples had detectable levels of IL-22 following

stimulation. It is also important to note that our cytokine assay only detects secreted cytokines present after the 44-h stimulation and does not account for cytokines consumed during the incubation period. In line with published literature, we show stimulation of PBMCs with SARS-CoV-2 spike proteins induces robust production of IFNγ and TNF[34,37]. Our overall findings of reduced cytokine secretion following XBB.1.5 spike stimulation of PBMCs suggest that the functional abilities of XBB.1.5 spike-specific T cells are attenuated compared to ancestral and BA.2 spike-specific T cells, possibly due to decreases in both the quantity and quality of antigen presentation. Nevertheless, the high frequency of circulating XBB.1.5 spike-reactive T cells in this cohort highlights the fact that the XBB.1.5 spike protein has not substantially evaded recognition by cellular immunity. Therefore, while the XBB.1.5 spike protein retains enhanced human receptor engagement and antibody evasion, pre-existing cellular immunity is expected to offer some protection upon exposure to the XBB.1.5 variant of SARS-CoV-2.

## Methods

This research complies with all relevant ethical regulations as approved by The University of British Columbia Clinical Research Ethics Board. Informed consent was obtained from all donors. There was no compensation for participation.

### Cloning, expression and purification of recombinant spike protein constructs

The production of the SARS-CoV-2 wild-type (D614G) and BA.2 HexaPro S proteins and RBDs and human ACE2 (residues 1–615) and human ACE2-FC (residues 1–740 with an FC tag), and mouse ACE2 (residues 1–615), and antibody Fabs (S309, S2M11, 4A8, 4–8, VH-FC ab8) were described previously[5,12,38].

The XBB.1.5 HexaPro S protein gene was synthesized and inserted into pcDNA3.1 (GeneArt Gene Synthesis, Thermo Fisher Scientific). PCR was used to amplify the XBB.1.5 RBD (amino acids 319–541) which was introduced in frame to the mu phosphatase signal sequence and incorporated within pcDNA3.1 via Gibson assembly (NEBuilder HiFi DNA Assembly Cloning Kit, New England Biolabs).

Expi293F cells were transiently transfected at a density of $3 \times 10^6$ cells/mL using linear polyethylenimine (Polysciences Cat# 23966-1). The media was supplemented 24 h after transfection with 2.2 mM valproic acid, and expression was carried out for 3–5 days at 37 °C, 8% $CO_2$. The supernatant was harvested by centrifugation and filtered through a 0.22-μm filter prior to purification.

For the XBB.1.5 ectodomain, supernatant was loaded onto a 5 mL HisTrap excel column (Cytiva). The column was washed for 20 CVs with wash buffer (20 mM Tris pH 8.0, 500 mM NaCl), 5 CVs of wash buffer supplemented with 20 mM imidazole, and the protein eluted with elution buffer (20 mM Tris pH 8.0, 500 mM NaCl, 500 mM imidazole). Elution fractions containing the protein were pooled and concentrated (Amicon Ultra 100 kDa cut off, Millipore Sigma) for gel filtration. Gel filtration was conducted using a Superose 6 10/300 GL column (Cytiva) pre-equilibrated with GF buffer (20 mM Tris pH 8.0, 150 mM NaCl). Peak fractions corresponding to soluble protein were pooled and concentrated (Amicon Ultra 100 kDa cut off, Millipore Sigma) Protein samples were flash-frozen in liquid nitrogen and stored at −80 °C.

For the XBB.1.5 RBD, supernatant was incubated with Ni-NTA resin (Qiagen) at 4 °C overnight. The resin was washed three times with 20 mM Tris pH 8.0, 500 mM NaCl, then three times with 20 mM Tris pH 8.0, 500 mM NaCl, supplemented with 20 mM of imidazole. Proteins were eluted in 20 mM Tris pH 8.0, 500 mM NaCl, containing 300 mM of imidazole and then subjected to Gel filtration using a Superose 6 10/ 300 GL column (Cytiva) pre-equilibrated with GF buffer (20 mM Tris pH 8.0, 150 mM NaCl). Peak fractions corresponding to soluble protein

were pooled and concentrated (Amicon Ultra 10 kDa cut off, Millipore Sigma) before flash freezing and storage at −80 °C.

### Biolayer Interferometry

BLI experiments were performed on the GatorBio plus instrument. Human ACE2-FC was immobilized onto protein A sensors prior to incubation in PBS to establish baselines. Concentrations (300 nM, 100 nM, 33 nM, 11 nM) of various RBD proteins were then incubated with the probes to allow association prior to incubation in PBS to for dissociation to occur. Curves were fit to a 1:1 binding model using the GatorBio evaluation software.

### Surface plasmon resonance

SPR experiments were performed on the Biacore T200 instrument. Human ACE2-FC was immobilized using the series S protein A chip. Increasing concentrations (6.25 nM, 31.25 nM, 62.5 nM, 125 nM, 250 nM) of various RBDs were injected over the surface for single cycle kinetic experiments. The surface was regenerated in 10 mM glycine pH 2.5. The experiments were performed at 25 degrees Celsius, using 10 mM HEPES pH 7.4, 150 mM NaCl, 3 mM EDTA and 0.05% v/v Surfactant P20 as running buffer. Reference-subtracted curves were fitted to a 1:1 binding model using Biacore evaluation software

### Monoclonal antibody enzyme linked immunosorbent assay

100 μL of wild-type (D614G), BA.2, or XBB.1.5 SARS-CoV-2 S proteins were coated onto 96-well MaxiSorp plates at 1 μg/mL in PBS overnight at 4 °C. All washing steps were performed three times with PBS + 0.05% Tween 20 (PBS-T). After washing, wells were incubated with blocking buffer (PBS-T + 2% casein) for 1 hour at room temperature. After washing, wells were incubated with dilutions of primary antibodies in PBS-T + 0.5% casein buffer for 1 hour at room temperature. After washing, wells were incubated with goat anti-human IgG (Jackson ImmunoResearch) at a 1:5,000 dilution in PBS-T + 2% casein buffer for 1 hour at room temperature. After washing, the substrate solution (Pierce 1-Step) was used for color development according to the manufacturer's specifications. Optical density at 450 nm was read on a Varioskan Lux plate reader (Thermo Fisher Scientific).

### Polyclonal antibody enzyme linked immunosorbent assay

In total, 100 μL of wild-type (D614G), BA.2, or XBB.1.5 SARS-CoV-2 S proteins were coated onto 96-well MaxiSorp plates at 1 μg/mL in PBS overnight at 4 °C. All washing steps were performed three times with PBS + 0.05% Tween 20 (PBS-T). After washing, wells were incubated with blocking buffer (PBS-T + 2% casein) for 1 h at room temperature. After washing, wells were incubated with dilutions of serum or purified bulk IgGs or Fabs in blocking buffer for 1 h at room temperature. After washing, wells were incubated with goat anti-human IgG (Jackson ImmunoResearch) at a 1:5000 dilution in PBS-T + 2% casein buffer for 1 h at room temperature. After washing, the substrate solution (Pierce 1-Step) was used for color development according to the manufacturer's specifications. Optical density at 450 nm was read on a Varioskan Lux plate reader (Thermo Fisher Scientific).

### Pseudovirus neutralization assay

Pseudoviruses harbouring a luciferase reporter gene and possessing WT, BA.2, and XBB.1.5 spike proteins were purchased from ProSci (cat # 95-200, 95-202, 95-208). HEK293T-ACE2-TMPRSS2 cells (BEI Resources cat# NR-55293) were seeded in 384-well plates at 20,000 cells per well. The next day, pseudovirus preparations were mixed with dilutions of sera or media alone prior to addition to cells and incubation for 72 h. Cells were then lysed and luciferase activity assessed using the ONE-Glo EX Luciferase Assay System (Promega) according to the manufacturer's specifications. Detection of relative luciferase units was carried out using a Varioskan Lux plate reader (Thermo Fisher).

Percent neutralization was calculated relative to signals obtained in the presence of virus alone.

## Mouse ACE2 enzyme linked immunosorbent assay

100 μL of mouse ACE2 was coated onto 96-well MaxiSorp plates at 5 μg/mL in PBS overnight at 4 °C. All washing steps were performed three times with PBS + 0.05% Tween 20 (PBS-T). After washing, wells were incubated with blocking buffer (PBS-T + 2% casein) for 1 h at room temperature. After washing, wells were incubated with dilutions of spike proteins in blocking buffer for 1 hour at room temperature. After washing, wells were incubated with mouse anti strep tag antibody (BIO-RAD Cat# MCA2489) at a 1:500 dilution in blocking buffer for 1 h at room temperature. After washing, wells were incubated with Goat Anti-Mouse IgG Fc Secondary Antibody, HRP (Invitrogen) at a 1:5,000 dilution in PBS-T + 2% casein buffer for 1 h at room temperature. After washing, the substrate solution (Pierce 1-Step) was used for color development according to the manufacturer's specifications. Optical density at 450 nm was read on a Varioskan Lux plate reader (Thermo Fisher Scientific).

## Generation of IgG and Fab fragments

First bulk IgG was isolated from sera as follows: sera was diluted 5 times in PBS before incubation with Protein A Agarose (Thermo Fisher) for 1 hour at room temperature. After washing the resin with 5 column volumes (CVs) of PBS IgG was eluted batchwise with 100 mM Glycine pH 3.5 immediately into 1/10th of the elution volume of 1 M Tris pH 8.0. IgGs were then either cleaved into Fabs or subjected to gel filtration using a Superose 6 10/300 GL column (Cytiva) pre-equilibrated with PBS. Peak fractions were pooled and concentrated (Amicon 100 kDa cut off, Millipore Sigma) before storage at 4 °C in 0.002% sodium azide.

For Fab cleavage, papin-agarose (Sigma) was added to the protein A IgG elutions and the mixture was supplemented to 10 mM EDTA and 10 mM L-Cystine. After incubation at 37° overnight, the reaction was centrifuged to remove the papain–agarose. The supernatant was then added to Protein A agarose resin and allowed to incubate for 20 min to an hour at room temperature before the flow through was collected and the resin washed with 3 CVs. The wash and flow through fractions were pooled and Fabs were concentrated (Amicon Ultra 10 kDa cut off, Millipore Sigma) before gel filtration using a Superose 6 10/300 GL column (Cytiva) pre-equilibrated with PBS. Peak fractions were pooled and concentrated (Amicon 10 kDa cut off, Millipore Sigma) before storage at 4 °C in 0.002% sodium azide.

## Generation of immune complexes for negative stain electron microscopy

For experiments using Fab fragments, 60 micrograms of WT or XBB.1.5 spike ectodomain was incubated with 3 mg of bulk Fabs in 300 μl of PBS overnight at room temperature. Size exclusion chromatography was then performed using a Superose 6 10/300 GL column (Cytiva) in PBS. Peak fractions were pooled and concentrated (Amicon 10 kDa cut off, Millipore Sigma) before negative stain electron microscopy analysis. For IgG experiments, 90 μg of XBB.1.5 spike ectodomain was incubated with 3.4 mg of bulk IgG in 300 μl of PBS overnight at room temperature. Size exclusion chromatography was then performed using a Superose 6 10/300 GL column (Cytiva) in PBS. Peak fractions were pooled and concentrated (Amicon 100 kDa cut off, Millipore Sigma) before negative stain electron microscopy analysis.

## CryoEM

For cryo-EM, 2.25 mg/mL S protein was vitrified on Quantifoil R1.2/1.3 Cu mesh 300 holey carbon grids after a glow discharge of 15 s at 10 mA. For S protein-hACE2 complex (1:1.2 S protein trimer: hACE2 molar ratio) and S protein-mACE2 complex (1:2.1 S protein trimer: mACE2 molar ratio), 2.25 mg/mL mixtures were vitrified on Quantifoil R1.2/1.3 Cu mesh 200 holey carbon grids, coated with 25 nm gold on each side, after a glow discharge of 20 s at 15 mA. All grids were glow discharged using a Pelco easiGlow glow discharge unit (Ted Pella) before 1.8 μL of protein suspension was applied to the surface of the grid at a temperature of 10 °C and a humidity level of >98%. Grids were subsequently blotted (12 s, blot force −10) and plunge frozen into liquid ethane using a Vitrobot Mark IV (ThermoFisher Scientific) plunge freezing device. Grids were imaged using a 300 kV Titan Krios G4 transmission electron microscope (TEM) (Thermo Fisher Scientific) equipped with a Falcon4 direct electron detector in electron event registration (EER) mode. Movies were collected at 155,000× magnification (calibrated pixel size of 0.5 Å per physical pixel) over a defocus range of −0.5 μm to −2 μm with a total dose of 40 e⁻/Å² using EPU automated acquisition software (ThermoFisher Scientific).

## CryoEM image processing

The detailed data processing workflow is illustrated in Supplementary Figs. S1–3. All data processing was performed in cryoSPARC v4. Motion correction in patch mode (EER upsampling factor 1, EER number of fractions 40), CTF estimation in patch mode, blob particle picking, and particle extraction (box size 400 Å) were performed in real time in cryoSPARC live. Multiple rounds of 3D and/or 2D classification were used to clean the particles. The final homogeneous refinement was performed with per particle CTF estimation and aberration correction.

For complexes of spike protein with human ACE2, particles were selected from second round of 3D classification, refined with C3 symmetry, symmetry-expanded and classified without alignment. Then local refinement was performed with a soft mask covering a single RBD and its bound ACE2.

For complexes of spike protein with mouse ACE2, particles were selected from second round of 3D classification, refined and classified without alignment. Then local refinement was performed with a soft mask covering a single RBD and its bound ACE2.

## Model Building and Refinement

For models of spike protein ectodomain alone, the SARS-CoV-2 BA.2 spike structure (PDB ID 8DM1) was used as initial model and docked into the map. Then, mutation and manual adjustment were performed in COOT, followed by iterative rounds of real space refinement in COOT and Phenix. For models of spike–ACE2 complex, the subcomplexes RBD–ACE2 were built using known RBD–ACE2 structures (PDB ID 8DM6 or 8DM8) as initial model and refined against local refinement maps. The resulting models were then docked into global refinement maps together with the other individual domains of the spike protein. Model was validated using MolProbity. Figures were prepared using UCSF Chimera, UCSF ChimeraX, and PyMOL (Schrodinger, LLC).

## Negative stain epitope mapping

SARS-CoV-2 spike (either WT or XBB.1.5)–Fab/IgG mixtures were diluted to -80 μg/ml in PBS and deposited onto grids (copper 300 mesh coated with mountainous ultrathin carbon) which had been plasma cleaned. Specimens were stained by 3 successive applications of 2% (w/v) uranyl formate (20 s, 20 s, 60 s). Negative stain grids were imaged using 200-kV Glacios (Thermo Fisher Scientific) TEMs equipped with Falcon3 and Falcon4 cameras operated in linear mode and EER mode, respectively. Micrographs were collected using EPU at nominal magnifications of 92,000× (physical pixel size 1.6 Å) and 120,000× (physical pixel size 1.2 Å) over a defocus range of −2.0 μm to −1.0 μm with a total accumulated dose of 40 and 60 e⁻/Å², respectively. Data processing was performed in CryoSPARC v4.0.1. Patch motion correction was performed, constant CTF values were output upon movie import. Blob picking was used, followed by 2D classification to generate templates for subsequent template picking. Several rounds of 2D classification on selected particles were then performed,

followed by ab-inito reconstructions, and either homogenous or heterogenous refinement to obtain the several reconstructions reported.

## T cell activation induced marker (AIM) assays

Whole heparinized blood was obtained from healthy donors via venipuncture. PBMCs were isolated by centrifugation of blood over Ficoll and stored in liquid nitrogen. Prior to stimulation, PBMCs were thawed and rested overnight in Immunocult media (StemCell Technologies) at $0.5-1 \times 10^6$ cells/mL in a 96-well plate. Cells were stimulated with 2 µg/ml of ancestral, BA.2, or XBB.1.5 spike protein or left unstimulated. Equal volumes of protein buffer (20 mM Tris pH8 150 mM NaCl) were added to each unstimulated condition. Two microlitres of anti-CD137-APC mAb was added to each well at the time of stimulation. PBMCs were incubated at 37 °C (5% $CO_2$, humidified atmosphere) for 44-h. Following stimulation, cells were centrifuged, and supernatant was collected and stored at −80 °C. Cell pellets were washed and resuspended in PBS and stained for 20 min with a cocktail of the antibodies in Supplementary Table 3. Following staining, cells were then washed twice with PBS, resuspended in 0.5% paraformaldehyde in PBS, and acquired on a 5-laser Symphony flow cytometer (BD Biosciences). A minimum of 100,000 events were acquired for each condition. Flow cytometry data analysis was performed using FlowJo v10.8.1 (BD Biosciences).

Cytokine analysis of PBMC supernatants was performed using LEGENDplex human Th Cytokine Panel V2 (Biolegend) following manufacturer's protocol, where samples were diluted at a 1:1 ratio in sample dilution buffer. Unstimulated cytokine readings were subtracted from stimulated conditions for each donor.

## Reporting summary

Further information on research design is available in the Nature Portfolio Reporting Summary linked to this article.

## Data availability

The data that support this study are available in the article and its Supplementary files or from the corresponding author upon request. The atomic models and cryo-EM density maps have been deposited into the Protein Data Bank (PDB) and Electron Microscopy Data Bank (EMDB) as follows: XBB.1.5 spike protein (PDB ID: 8VKK, EMDB ID: EMD-43320) mACE2 x2 + XBB.1.5 S (PDB ID: 8VKL, EMDB ID: EMD-43321). mACE2 + XBB.1.5 S (PDB ID: 8VKM EMDB ID: EMD-43322). Focused refined mACE2 + XBB1.5 RBD (PDB ID: 8VKN, EMDB ID: EMD-43323). hACE2x3 + XBB.1.5 S (PDB ID: 8VKO, EMDB ID: EMD-43324). Focused refined hACE2 + XBB1.5 RBD (PDB ID: 8VKP, EMDB ID: EMD-43325). The negative stain EM density maps of antibody-spike complexes derived from donor serum are deposited together under the following EMDB entry: Negative Stain EM Reconstructions of SARS-CoV-2 spike proteins mixed with polyclonal antibodies from donor 4 (EMDB ID: EMD-43326) Source data are provided with this paper.

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

## Acknowledgements

We thank the study participants. This work was supported by awards to S.S. from a Canada Excellence Research Chair Award; the VGH Foundation; Genome BC, Canada; and the Tai Hung Fai Charitable Foundation. D.M. is supported by a CIHR Vanier Canada Graduate Scholarship. J.W.S. is supported by a CIHR Frederick Banting and Charles Best Canada Graduate Scholarships Doctoral Award (CGS-D) and a University of British Columbia President's Academic Excellence Initiative PhD Award. C.P. is supported by a University of British Columbia Four Year Doctoral Fellowship.

## Author contributions

D.M., J.W.S., K.T., and S.A. carried out expression and purification of all proteins. D.M., J.W.S., F.V., and K.T. performed ELISA experiments. D.M. and J.W.S. performed neutralization experiments. D.M., J.W.S., and S.C. performed SPR and BLI experiments. C.P. performed the T cell experiments under supervision of T.S. and with input from L.C.. A.B. and K.S.T. carried out the experimental components of cryo-EM and electron microscopy including specimen preparation and data collection. X.Z. and D.M. carried out all computational aspects of image processing and structure determination. D.M., J.W.S., X.Z., and S.S. interpreted and analyzed the cryo-EM structures. D.M., J.W.S., C.P. and S.S. drafted the initial manuscript with input from all authors.

## Competing interests

S.S. is the Founder and CEO of Gandeeva Therapeutics Inc. T.S. has previously maintained a research contract with AbCellera Biologics Inc. The remaining authors declare no competing interests.
