## [Peer Review File · Nature Communications]

SARS-CoV-2 XBB.1.5 Spike Protein: Altered Receptor Binding, Antibody Evasion, and Retention of T Cell RecognitionREVIEWER COMMENTS

Reviewer #1 (Remarks to the Author):

The manuscript present receptor binding studies of Spike from XBB.1.5 to ACE2. The novel finding is that binding of XBB.1.5 Spike to human ACE2 is preserved as compared to previous Omicron variants, but that there is a decrease in XBB.1.5 Spike binding to murine ACE2. Further, the results from this study confirm previous data on evasion of neutralizing antibodies binding the different Omicron variants, and the broad reactivity of T-cell responses raised against SARS-CoV-2.

There is a large body of evidence in scientific literature on the different Omicron variants with respect to evading previously formed neutralizing antibodies(e.g. Cao et al, 2021, Nature; McCallum et al, 2022, Science). It has also been shown that the F486P mutation of XBB.1.5 enhances ACE2 binding (e.g. Qu et al, 2023, Cell Reports). Fortunately, previous studies also demonstrate the maintenance of protection against severe disease, likely due to cellular immunity (e.g. Buchan et al, 2022 JAMA Netw Open; Feikin et al, 2022, Vaccine). Previously, T-cell responses formed after vaccination and infection have been demonstrated responsive to a wide range of coronavirus (e.g. Liu et al, 2022, Nature; Grifoni et al, 2020, Cell). In accordance with these data, the present study confirms that cellular immunity is also preserved against the XBB.1.5 variant.

The study is methodologically sound, and experiments seem reproducible and reliable. It is a strength that different methods have been employed to test validity of conclusions from different angles (e.g. the use of SPR, cryoEM, and ELISA for evaluating binding to ACE2). As for evaluation of antibody binding to Spike, the authors first evaluated the ability of a small panel of mAbs to bind Spike of XBB.1.5. Only one of these bound, and when progressing to testing of sera from human donors they indeed saw binding to XBB.1.5 (albeit reduced as compared to WT, as expected). It is interesting that the authors found pre-existing IgG capable of binding also XBB.1.5. When selecting a donor to see if a conserved binding site could be identified, the authors identified an ACE2 competitive binding site in the RBD. As such, a capacity for neutralization is indicated, but there are indeed several effector mechanisms besides neutralization that could also be of importance for viral clearance.

Reviewer #2 (Remarks to the Author):

The manuscript entitled “SARS-CoV-2 XBB.1.5 Spike Protein: Altered Receptor Binding, Antibody Evasion, and Retention of T Cell Recognition” by Mannar et al characterizes the interaction between XBB.1.5 spike with human and mouse ACE2 and the humoral and T cell immunity from SARS-CoV-2 (+) sera. The data consist of biochemical analysis of variant spike proteins with ACE2 interactions using cryo-EM surface plasmon resonance, and ELISA. Further, the study examined monoclonal antibodies and polyclonal human anti-SARS-CoV-2 sera interactions with variant BA.2 and XBB.1.5 spike proteins to evaluate its potential to provide humoral protection against the current circulating SARS-CoV-2 VOC. Further, the manuscript describes the analysis of the T cell response against SARS-CoV-2 WT, BA.2, and XBB.1.5 from subjects. Overall, this is a significant and important study characterizing the molecular interaction of XBB.1.5 spike with ACE2 and the human SARS-CoV-2 immunity against the XBB.1.5 VoC.

The biochemical studies provide molecular insight into the XBB.1.5 spike/human ACE2 interactions and propose a plausible reason of the enhanced interaction of BA.2 and XBB.1.5 spike with ACE2. The analysis of mouse ACE2 with XBB.1.5 spike seems a little out of place. Hence, the authors should include more rationale and significance of the mouse ACE2-XBB.1.5 interaction studies.

The analysis of the polyclonal anti-SARS-CoV-2 sera interactions is experimentally sound. However, the authors should examine if the sera can neutralize the intact virus or a pseudovirus containing XBB.1.5 because binding does not always translate into virus neutralization or protection against virus. Further, the T cell analysis using cells from SARS-CoV-2 immunized/infected individuals demonstrates a Th1 T cell response to XBB.1.5 and decrease in some cytokines. These studies would be more impactful if these responses were based on S1 or S2 domains of spike.

Other minor points that should be addressed:

1) Figure 1A color coding is not explained. Is the shaded box the mutation?

2) The manuscript jumps from Figure 2 to Figure 4. This is atypical for describing data. Also, text refers to Figure 4G, which does not exist.

Reviewer #3 (Remarks to the Author):

In the report “SARS-CoV-2 XBB.1.5 Spike Protein: Altered Receptor Binding, Antibody Evasion, and Retention of T Cell Recognition” by Mannar, Saville, Poloni et al., the authors report the results of a study designed to investigate the molecular mechanism of SARS-CoV-2 immune evasion. While the results of the study could be of general interest to the field, the below-detailed points decrease reviewer enthusiasm for the report in its current form.

With respect to figure 2A, the SPR profiles of the WT and BA.2 protein seem very similar, but the reported KDs are approximately 10-fold different. In contrast, the binding profiles of BA.2 and XBB.1.5 seem very different, but the reported KDs are almost identical. Can the authors explain this? Moreover, if the authors want to make an argument about the similarities/differences in KD values, they would need to run the experiment three or more times, determine KDs for each run and perform a statistical analysis of the resulting findings.

A few figure panels are incorrectly referenced in the body of the report (e.g., 2G is called 4G).

The reproducibility of the results presented in figure 3A are unclear. The authors should report binding results obtained across three or more independent experiments and perform a statistical analysis of the resulting data.

Based on the two cryo-EM structures, the authors make multiple suggestions as to why the binding of the various version of the spike protein to mouse and human ACE2 may be different. However, the authors do not test any of these suggestions. It seems it would greatly increase the potential impact of the work if the authors made a select set of single point mutations focused on these residues of interest and tested the impact of these mutations on the interaction of the spike with ACE2. Without this follow-up study, the suggestions drawn from the cryo-EM studies lack support.

The reproducibility of the findings shown in figure 4A and 4B are unclear. There is also no statistical analysis of the resulting findings in 4A.

Related to figure 4C, the authors failed to find spike-fab densities from samples of XBB.1.5 and human immune serum and draw the conclusion that the serum does not have 1.5-reactive antibodies. But how can the authors demonstrate that this was not simply a technical problem with this sample? What is the positive control that demonstrates that IF the antibodies were present, they would have been detected in that experimental run?

Related to figure 4D, the reproducibility of the reported findings is again unclear. Moreover, if the authors want to draw conclusions about the similarity/difference between the binding of various antibody preparations to different spike proteins, they would need to determine EC50 values across three or more independent experiments and then run a statistical analysis of the resulting data.

Pages 12-13 – The sentence “Thus, this finding highlights the ability for pre-existing serum IgG molecules to recognize the XBB.1.5 RBD at neutralising epitopes.” Should be modified to “Thus, this finding highlights the ability for pre-existing serum IgG molecules to recognize the XBB.1.5 RBD at potentially neutralising epitopes.” as the authors did not directly measure the neutralizing ability of the antibodies under consideration.

The results presented in figure 5 are a survey of the T cell reactivity from a series of SARS-CoV-2 immune donors. As such, it seems to add little to the report. At a minimum, it would be interesting to know if there is any correlation between donor status (e.g., vaccination regiment) and the pattern of T cell reactivity. There is also no discussion of how the level of any particular subset of T cells in a particular donor do (or do not) correlate with the levels of the various cytokines analyzed. There is also no discussion of the HLA haplotype of each donor and if this may impact the findings. This aspect of the report seems underdeveloped.

Reviewer #4 (Remarks to the Author):

The manuscript presented by Mannar and coworkers provides a thorough analysis of the emerging XBB.1.5 variant of SARS-CoV-2, which has gained global prevalence. The study employs cryo-electron microscopy (cryo-EM) to elucidate the structural features of the XBB.1.5 spike protein highlighting the interactions with both human and mouse ACE2 receptors. Additionally, the study underscores the XBB.1.5 variant's ability to evade monoclonal and polyclonal antibodies, while retaining some binding capability with pre-existing serum IgG molecules. The study also examines XBB.1.5-specific CD8⁺ and CD4⁺ T cell responses, revealing a skewed Th1 phenotype and attenuated cytokine secretion.

The methodology, data interpretation and conclusions appear robust and valid. Even though there are recent reports on the XBB.1.5 variant, the current manuscript offers key findings and contributes to the understanding of its virological characteristics and its interactions with the immune system.

Overall, the manuscript's clarity, context, and methodological rigor contribute to its potential significance. However, providing more explicit connections to recent relevant literature could enhance the manuscript's contextualization.

Major points:

1. The authors describe the cryo-EM structure of the XBB.1.5 spike ectodomain with an overall architecture that shows similarity with previous variants. Tamura et al. has recently determined the structures of XBB.1 S ectodomain alone and the XBB.1 S-ACE2 complex by cryoelectron microscopy analysis. The reference of this paper is missing, and the authors should expand this section by providing a more detailed description on the structural features of XBB.1.5 spike protein. Alternatively, the authors could consider to combine the first two results by describing the structural and biochemical features of XBB.1.5 Spike with ACE2 receptor.

2. Other reports have demonstrated enhanced receptor binding by the XBB.1.5 spike protein compared to XBB.1 variant, indicating a higher affinity of XBB.1.5 for the ACE2 receptor with KD values of 3,4nM and 19nM, respectively (Yue et al. 2023). Yeast surface

display assay showed that the dissociation constant value of XBB.1.5 S receptor-binding domain from the human ACE2 receptor is significantly (4.3 times) lower than that of XBB.1 S receptor-binding domain (Uriu et al. 2023). In the current manuscript, the authors observed a strong binding affinity of XBB.1.5 RBD to hACE2 receptor with K_D 0,75nM. How do the authors rationalize discrepancies in nM affinity through consistent methodology?

3. The authors describe that the affinity of the XBB.1.5 spike for hACE2 is enhanced to a similar extent as BA.2, when compared to WT. With that regards, it would be preferable to include the XBB.1 RBD in the surface plasmon resonance (SPR) experiments. By doing so, the authors could gain a clearer and more comprehensive understanding of the described enhanced binding affinity for the XBB.1.5 spike proteins compared to the XBB.1 variant, as well as the contribution of the S486P mutation.

4. The authors well explain the structural basis of the receptor binding by cryo-EM studies of XBB.1.5 in complex with hACE2. Zhang et al. have recently conducted crystallographic studies on the receptor-binding domains (RBDs) from the subvariants XBB.1 and XBB.1.5, each complexed with human ACE2. Does the cryo-EM and crystal-based models superimpose well or there is a structural variation? This additional comparison would provide valuable insights into the interactions observed, i.e., the favorable hydrophobic interaction with M82 and L79. The study indeed underscores phenylalanine as the best adapted to hACE2 at position 486, followed by a proline and then a serine.

5. The authors demonstrate enhanced retention of S309 binding to the XBB.1.5 spike protein - as compared to BA.2 - potentially due to the differential G339D (BA.2) versus G339H (XBB.1.5) mutation, which occurs within the S309 epitope. Cao et al. highlighted that, even though D339H is a charge-reversing mutation on the S309 binding interface, the neutralizing activity of S309 was not affected and even exhibited slightly improved neutralization activity against BA.2.75 compared with BA.2 and BA.4/5. Recently, Qu and coworker also showed enhanced resistance of the sensitivity by XBB variants to neutralization by S309, highlighting the role of R346, G339H and L368I mutations occurred in XBB.1.5 spike variant. The authors should include this recent work in the discussion.

Other comments:

1. The phrase "sought to" appears frequently in the manuscript; please replace it with more dynamic verbs.
2. The major findings outlined in the discussion, listed from 1 to 4, could be presented in more narrative fashion.
3. Receptor binding studies included in the manuscript demonstrate the retention of binding contacts with the human ACE2 receptor by XBB.1.5 variant. In the discussion, the authors summaries this result as increased XBB.1.5 spike - hACE2 affinity as compared to WT spike and similar affinity as compared to other Omicron variant spikes. To align with this conclusion the authors might consider rephrasing result's title "Enhanced Binding of Human ACE2 by the XBB.1.5 Spike Protein", thus enhancing the title coherence with the study's content.

Reviewer #1

The manuscript presents receptor binding studies of Spike from XBB.1.5 to ACE2. The novel finding is that binding of XBB.1.5 Spike to human ACE2 is preserved as compared to previous Omicron variants, but that there is a decrease in XBB.1.5 Spike binding to murine ACE2. Further, the results from this study confirm previous data on evasion of neutralizing antibodies binding the different Omicron variants, and the broad reactivity of T-cell responses raised against SARS-CoV-2.

There is a large body of evidence in scientific literature on the different Omicron variants with respect to evading previously formed neutralizing antibodies (e.g. Cao et al, 2021, Nature; McCallum et al, 2022, Science). It has also been shown that the F486P mutation of XBB.1.5 enhances ACE2 binding (e.g. Qu et al, 2023, Cell Reports). Fortunately, previous studies also demonstrate the maintenance of protection against severe disease, likely due to cellular immunity (e.g. Buchan et al, 2022 JAMA Netw Open; Feikin et al, 2022, Vaccine). Previously, T-cell responses formed after vaccination and infection have been demonstrated responsive to a wide range of coronavirus (e.g. Liu et al, 2022, Nature; Grifoni et al, 2020, Cell). In accordance with these data, the present study confirms that cellular immunity is also preserved against the XBB.1.5 variant.

The study is methodologically sound, and experiments seem reproducible and reliable. It is a strength that different methods have been employed to test validity of conclusions from different angles (e.g. the use of SPR, cryoEM, and ELISA for evaluating binding to ACE2). As for evaluation of antibody binding to Spike, the authors first evaluated the ability of a small panel of mAbs to bind Spike of XBB.1.5. Only one of these bound, and when progressing to testing of sera from human donors they indeed saw binding to XBB.1.5 (albeit reduced as compared to WT, as expected). It is interesting that the authors found pre-existing IgG capable of binding also XBB.1.5. When selecting a donor to see if a conserved binding site could be identified, the authors identified an ACE2 competitive binding site in the RBD. As such, a capacity for neutralization is indicated, but there are indeed several effector mechanisms besides neutralization that could also be of importance for viral clearance.

We thank the reviewer for their thorough reading of our manuscript and contextualization within the literature. We have updated the discussion section to include the potential effector mechanisms that may be mediated by antibodies.

Reviewer #2

The manuscript entitled “SARS-CoV-2 XBB.1.5 Spike Protein: Altered Receptor Binding, Antibody Evasion, and Retention of T Cell Recognition” by Mannar et al characterizes the interaction between XBB.1.5 spike with human and mouse ACE2 and the humoral and T cell immunity from SARS-CoV-2 (+) sera. The data consist of biochemical analysis of variant spike proteins with ACE2 interactions using cryo-EM surface plasmon resonance, and ELISA. Further, the study examined monoclonal antibodies and polyclonal human anti-SARS-CoV-2 sera interactions with variant BA.2 and XBB.1.5 spike proteins to evaluate its potential to provide humoral protection against the current circulating SARS-CoV-2 VOC. Further, the manuscript describes the analysis of the T cell response against SARS-CoV-2 WT, BA.2, and XBB.1.5 from subjects. Overall, this is a significant and important study characterizing the molecular interaction of XBB.1.5 spike with ACE2 and the huma SARS-CoV-2 immunity against the XBB.1.5 VoC.

We thank the reviewer for their efforts in reviewing our manuscript and their enthusiasm for the significance and importance of the study.

The biochemical studies provide molecular insight into the XBB.1.5 spike/human ACE2 interactions and propose a plausible reason for the enhanced interaction of BA.2 and XBB.1.5 spike with ACE2. The analysis of mouse ACE2 with XBB.1.5 spike seems a little out of place. Hence, the authors should include more rationale and significance of the mouse ACE2-XBB.1.5 interaction studies.

Thank you for raising this important point regarding the rationale for investigating mouse ACE2 binding. A prevalent hypothesis for the origin of the Omicron lineage variants is that they evolved within a mouse reservoir and then achieved zoonotic transmission back to humans. Consistent with this theory, the first two omicron lineages (BA.1 and BA.2) have spike proteins that uniquely possess high affinity for mouse ACE2, while all previous lineages did not. Thus our motivation for the mouse ACE2 experiments is to assess if this unique property has been preserved in the XBB.1.5 spike protein. We find that the propensity to bind mouse ACE2 with high affinity has been lost, with interesting implications for the mouse origin theory. This result is consistent with the binding to mouse ACE2 no longer representing a relevant selection pressure on the evolutionary trajectory of Omicron lineage spike proteins now that humans are the main reservoir.

We have added this additional rationale into the introduction for the mouse ACE2 - XBB.1.5 spike interaction study and summarize the significance of studying this interaction within the results and discussion section.

The analysis of the polyclonal anti-SARS-CoV-2 sera interactions is experimentally sound. However, the authors should examine if the sera can neutralize the intact virus or a pseudovirus containing XBB.1.5 because binding does not always translate into virus neutralization or protection against virus.

As requested, we have performed pseudovirus neutralization assays against wild-type, BA.2, and XBB.1.5 spike proteins with all sera samples examined in this study. These new results are summarized in Figure 4B and agree well with the ELISA binding experiments as shown below:

[Editorial Note: The citation for the table on this page is as follows: Poloni, C., Schonhofer, C., Ivison, S., Levings, M.K., Steiner, T.S. and Cook, L. (2023), T-cell activation–induced marker assays in health and disease. *Immunol Cell Biol*, 101: 491-503. <https://doi.org/10.1111/imcb.12636>]

Further, the T cell analysis using cells from SARS-CoV-2 immunized/infected individuals demonstrates a Th1 T cell response to XBB.1.5 and decrease in some cytokines. These studies would be more impactful if these responses were based on S1 or S2 domains of spike.

We agree with the reviewer that it would be interesting to study domain-specific (S1 vs S2) T cell responses in SARS-CoV-2 immunized/infected individuals. Our analysis of spike-specific T cells relies on whole spike protein for stimulation which, in addition to complete peptide pools, is standard practice in activation induced T cell marker assays (as summarized in table 1 Poloni, et al. 2023). Unfortunately, we do not have access to additional sera from the cohort reported here, and resampling would not be adequate to evaluate pre-existing XBB.1.5 cellular immunity, as the population has now been widely exposed to this variant, which was not the case when we began experimentation.

Table 1. Summary of studies using AIM assays to interrogate SARS-CoV-2-specific T-cell responses.

Main findings	Date published	AIM markers	Cell type	Stimulus	Duration of assay (h)	Reference
T cells recognize SARS-CoV-2 M, N and S peptides in 40–60% of unexposed individuals	May 2020	CD4 ⁺ : OX40 + 4-1BB CD8 ⁺ : CD69 + 4-1BB	PBMCs (1 × 10 ⁶ cells/well, 96-well plate)	Spike, matrix and nucleocapsid peptide megapool (1 µg mL ⁻¹)	24	⁹⁹
Common cold coronavirus-specific CD4 ⁺ T cells are cross-reactive to COVID-19	October 2020	CD4 ⁺ : OX40 + 4-1BB	PBMCs (1 × 10 ⁶ cells/well, 96-well plate)	Spike peptide megapool (1 µg mL ⁻¹)	24	¹⁰⁹
Coordinated T-cell and antibody responses associate with milder COVID-19, with impairments related to aging	November 2020	CD4 ⁺ : OX40 + 4-1BB + CD40L CD8 ⁺ : CD69 + 4-1BB	PBMCs (1 × 10 ⁶ cells/well, 96-well plate)	Spike peptide megapool (1 µg mL ⁻¹)	24	³²
Memory T cells persist for 3 months after mild COVID-19 infection	January 2021	CD4 ⁺ : ICOS + CD40L	PBMCs (4 × 10 ⁶ cells mL ⁻¹)	Full-length spike protein (2 µg mL ⁻¹)	20	¹¹⁰
SARS-CoV-2-specific Th1 and Tfh cell responses following first vaccination dose correlate with antibody levels after second dose	September 2021	CD4 ⁺ : CD200 + CD40L CD8 ⁺ : 4-1BB (+ IFN-γ)	Frozen PBMCs (1 × 10 ⁶ cells/well, 96-well plate), rested overnight	Spike peptide megapool (1 µg mL ⁻¹) + anti-CD28/CD49b	24	¹⁰¹
BNT126b2 vaccination induces SARS-CoV-2 omicron variant-specific CD4 ⁺ and CD8 ⁺ T cells	January 2022	CD4 ⁺ : CD69 + CD40L CD8 ⁺ : CD69 + CD40L	Frozen PBMCs (1 × 10 ⁶ /well, 96-well plate), rested for 4 h	15mer OR 20mer overlapping spike peptide pools (1 µg mL ⁻¹)	12	¹¹¹
Patients with COVID-19 have defective spike-specific Tfh responses following mRNA vaccination	February 2022	CD4 ⁺ : CD25, OX40	Fresh PBMCs (1 × 10 ⁶ cells mL ⁻¹), 24-well plate	Spike overlapping peptide pool (1 µg mL ⁻¹)	48	¹¹²
Vaccine-induced SARS-CoV-2 spike-specific T-cell responses to variant-derived peptides were not significantly different from the original Wuhan strain	March 2022	CD4 ⁺ : OX40 + 4-1BB CD8 ⁺ : CD69 + 4-1BB	Frozen PBMCs (1 × 10 ⁶ /well, 96-well plate), rested for 4 h	Spike 15mer overlapping peptide pool or megapool (1 µg mL ⁻¹)	20	¹⁰²
Previously infected, vaccinated individuals have a distinct subset of CD127 ^{low} IL-10 ⁺ type 1 regulatory T cells	April 2022	CD4 ⁺ : CD69 + 4-1BB or CD69 + CD40L	Frozen PBMCs (5 × 10 ⁶ cells mL ⁻¹), polystyrene tubes	SARS-CoV-2 S, M and N peptide megapools (5 µg mL ⁻¹)	18	¹¹³
Development of the B and T cell tandem lymphocyte evaluation (BATTLE) assay for concurrent detection of spike-specific B- and T cells	June 2022	CD4 ⁺ : CD69 + OX40 CD8 ⁺ : CD25 + CD69	Frozen PBMCs (5 × 10 ⁶ cells mL ⁻¹), polystyrene plates	Overlapping spike peptides (1 µg mL ⁻¹)	24	¹¹⁴
mRNA, Novavax and adenovirus-based SARS-CoV-2 vaccines induce spike-specific CD4 ⁺ and CD8 ⁺ T cells in most individuals 6 months after the first dose. mRNA and Novavax vaccines induce a greater magnitude of spike-specific CD4 ⁺ T cells than natural infection	July 2022	CD4 ⁺ : OX40 + CD40L + CD137 CD8 ⁺ : CD69 + CD137	Frozen PBMCs in 96-well plates	SARS-CoV-2 peptide megapool (1 µg mL ⁻¹)	24	¹¹⁵
SARS-CoV-2-specific CD4 ⁺ T-cell responses were reduced in preschool-aged children and increased with age	July 2022	CD4 ⁺ : CD69 + OX40	Frozen PBMCs in 96-well plates	SARS-CoV-2 and spike peptide megapools (1 µg mL ⁻¹)	24	¹¹⁶
T-cell immunity 7 months after low-dose mRNA vaccination is similar to infection-driven immunity	September 2022	CD4 ⁺ : OX40 + 4-1BB CD8 ⁺ : CD69 + 4-1BB	PBMCs (1 × 10 ⁶ cells/well, 96-well plates)	SARS-CoV-2 peptide megapool (1 µg mL ⁻¹)	24	¹⁰⁹

COVID-19, coronavirus disease 2019; IFN, interferon; IL, interleukin; mRNA, messenger RNA; PBMC, peripheral blood mononuclear cell; SARS-CoV-2, severe acute respiratory syndrome coronavirus 2; Tfh, T follicular helper; Th1, T helper type 1.

Other minor points that should be addressed:

1) Figure 1A color coding is not explained. Is the shaded box the mutation?

Yes, it is correct that shaded boxes are mutations and each color corresponds to a different lineage of SARS-CoV-2. We have updated the figure caption to reflect these missing details.

2) The manuscript jumps from Figure 2 to Figure 4. This is atypical for describing data. Also, text refers to Figure 4G, which does not exist.

We apologize for the confusion, the reference to Figure 4G should actually be to Figure 2G, and the manuscript no longer jumps from Figure 2 to Figure 4. Thank you for catching this error.

Reviewer #3

In the report “SARS-CoV-2 XBB.1.5 Spike Protein: Altered Receptor Binding, Antibody Evasion, and Retention of T Cell Recognition” by Mannar, Saville, Poloni et al., the authors report the results of a study designed to investigate the molecular mechanism of SARS-CoV-2 immune evasion. While the results of the study could be of general interest to the field, the below-detailed points decrease reviewer enthusiasm for the report in its current form.

We thank the reviewer for their time and effort in reviewing the manuscript and have provided a detailed response to their points below.

With respect to figure 2A, the SPR profiles of the WT and BA.2 protein seem very similar, but the reported KDs are approximately 10-fold different. In contrast, the binding profiles of BA.2 and XBB.1.5 seem very different, but the reported KDs are almost identical. Can the authors explain this? Moreover, if the authors want to make an argument about the similarities/differences in KD values, they would need to run the experiment three or more times, determine KDs for each run and perform a statistical analysis of the resulting findings.

The kinetic definition of KD refers to the ratio of the dissociation and association rates. Therefore there exist many combinations of association and dissociation rates that could all give rise to similar KD values. As such, association/dissociation curves that may visually appear different, may indeed be fit to estimate KD values that are similar by virtue of this ratio. While the curves for WT and BA.2 experiments may appear similar in Figure 2, there are obvious differences at the final dissociation phase of these curves, particularly in both the rate and extent of dissociation. It is clear that the dissociation occurs more rapidly and completely for the WT curve. Thus it is expected that the BA.2 data will be fit to estimate a higher affinity interaction, which is observed in the derived KDs.

With regards to the SPR data and replicates, we performed these experiments using 5 consecutive injections of a binding partner and derive the KD using data from all injections, which is a standard in the field. We therefore note that our data has appropriate technical replication, which matches the field standards.

To further validate our interpretation of the SPR data, we have performed an independent and complementary set of experiments using biolayer interferometry (BLI). Again, like SPR, each BLI run involves averaging the KD estimated from the collective fitting of data obtained using several concentrations of a binding partner, and is a standard in the field. We have nevertheless taken the reviewer’s suggestion to perform at least 3 experiments for each interaction and have summarized the results below, along with providing representative binding curves (all binding curves appear in Supplemental Figure 2 in the revised manuscript). These results are in full agreement with our

interpretation of the SPR data, primarily that both BA.2 and XBB.1.5 RBDs possess similarly enhanced ACE2 potency compared to the WT RBD. We have updated the manuscript with this additional data.

A few figure panels are incorrectly referenced in the body of the report (e.g., 2G is called 4G).

We apologize for the confusion and have ensured that all figure panels are correctly referenced in the revised manuscript.

The reproducibility of the results presented in figure 3A are unclear. The authors should report binding results obtained across three or more independent experiments and perform a statistical analysis of the resulting data.

We have clarified the technical and experimental replication performed for figure 3A in the figure caption: "The results of two independent experiments are plotted, with technical triplicates performed in each experiment." While we are not in a position to repeat these experiments a third time, we believe that the trend of WT<XBB.1.5<BA.2 in mACE2 binding affinity is apparent from these results.

Based on the two cryo-EM structures, the authors make multiple suggestions as to why the binding of the various version of the spike protein to mouse and human ACE2 may be different. However, the authors do not test any of these suggestions. It seems it would greatly increase the potential impact of the work if the authors made a select set of single point mutations focused on these residues of interest and tested the impact of these mutations on the interaction of the spike with ACE2. Without this follow-up study, the suggestions drawn from the cryo-EM studies lack support.

We thank the reviewer for this suggestion and agree that a mutational analysis would certainly provide additional data of relevance. However such a detailed analysis of these interactions is outside the scope of the current manuscript - which strives to report the structures of these binding interfaces and to generate hypotheses about specific mutational effects that can be characterized in-depth in follow-up studies.

The reproducibility of the findings shown in figure 4A and 4B are unclear. There is also no statistical analysis of the resulting findings in 4A.

We have clarified the reproducibility of these findings in the figure caption as follows: "The results are from a single experiment performed with technical triplicates." To prevent confusion about the type of data, we have opted to present the results in a table.

Related to figure 4C, the authors failed to find spike-fab densities from samples of XBB.1.5 and human immune serum and draw the conclusion that the serum does not have 1.5-reactive antibodies. But how can the authors demonstrate that this was not simply a technical problem with this sample? What is the positive control that demonstrates that IF the antibodies were present, they would have been detected in that experimental run?

To clarify, we do not draw the conclusion that the serum does not have XBB.1.5 reactive antibodies. Our interpretation of these findings includes: "Despite the significant loss of serum IgG potency, all serum samples exhibited binding of the XBB.1.5 spike protein, suggesting the presence of conserved epitopes within the XBB.1.5 spike protein which may be targeted by pre-existing serum antibodies." Our binding

analyses (Figure 4D) confirm the existence of these antibodies and serve as the basis for our experiments in Figure 4E-F.

The lack of spike fab-densities when using XBB.1.5 spike as compared to WT spike in the experiment shown in Figure 4C is most likely due to antigenic variation at the relevant epitopes described, leading to a reduction in binding affinity of these antibodies for the XBB.1.5 spike protein. The reviewer raises the possibility that our finding may be confounded by the absence of the antibody fragments due to a technical error. To this point, we performed incubations of the wild-type spike and the XBB.1.5 spike protein with an excess of polyclonal Fabs prior to size exclusion chromatography (SEC) to separate unbound fabs from spike-Fab complexes (as described in the methods section). Thus, we have direct biochemical confirmation that the antibody fragments in question were indeed present in the mixture as shown in the SEC traces below.

Related to figure 4D, the reproducibility of the reported findings is again unclear. Moreover, if the authors want to draw conclusions about the similarity/difference between the binding of various antibody preparations to different spike proteins, they would need to determine EC₅₀ values across three or more independent experiments and then run a statistical analysis of the resulting data.

We have clarified the reproducibility of the experiment in the figure caption: "Experiments were performed once in technical quadruplicate (n = 4) and are shown as points." We have additionally included error estimates in the derived EC₅₀ values. Unfortunately we have limited amounts of this antibody preparation and cannot perform additional experimental runs. These binding assays were hypothesis generating experiments that pointed towards the possibility of IgGs being able to cross react with the XBB.1.5 spike to a greater extent than Fab fragments by virtue of avidity. This finding prompted our structural studies presented in Figure 4E-F, which further supported this point.

Pages 12-13 – The sentence "Thus, this finding highlights the ability for pre-existing serum IgG molecules to recognize the XBB.1.5 RBD at neutralising epitopes." Should be modified to "Thus, this finding highlights the ability for pre-existing serum IgG molecules to recognize the XBB.1.5 RBD at potentially neutralising epitopes." as the authors did not directly measure the neutralizing ability of the antibodies under consideration.

We have revised this sentence as suggested.

The results presented in figure 5 are a survey of the T cell reactivity from a series of SARS-CoV-2 immune donors. As such, it seems to add little to the report. At a minimum, it would be interesting to know if there is any correlation between donor status (e.g., vaccination regiment) and the pattern of T cell reactivity. There is also no discussion of how the level of any particular subset of T cells in a particular donor do (or do not) correlate with the levels of the various cytokines analyzed. There is also no discussion of the HLA haplotype of each donor and if this may impact the findings. This aspect of the report seems underdeveloped.

The inclusion of the T cell data within this manuscript strives to further characterize the immune response against WT, BA.2, and XBB.1.5 spike proteins. While we observed no significant differences in overall CD4+ nor CD8+ activation across the variant spike proteins, we find this to be a pertinent negative result, that confirms preserved T cell responses in emerging SARS-CoV-2 variants. Further, the significantly decreased cytokine secretion from XBB.1.5 stimulated T cells does distinguish the XBB.1.5 spike from WT and BA.2, as discussed in the last paragraph of the discussion section. Unfortunately, our T cell cohort is small (10 donors) confounding any meaningful correlation analysis between T cell reactivity and vaccine regiment, especially given the diversity in vaccination identity, number of doses, and dose timeline (Table S2). We agree with the reviewer that an analysis of HLA haplotype for each donor would be interesting, however we unfortunately lack additional sample to measure this aspect of cellular immunity.

Reviewer #4

The manuscript presented by Mannar and coworkers provides a thorough analysis of the emerging XBB.1.5 variant of SARS-CoV-2, which has gained global prevalence. The study employs cryo-electron microscopy (cryo-EM) to elucidate the structural features of the XBB.1.5 spike protein highlighting the interactions with both human and mouse ACE2 receptors. Additionally, the study underscores the XBB.1.5 variant's ability to evade monoclonal and polyclonal antibodies, while retaining some binding capability with pre-existing serum IgG molecules. The study also examines XBB.1.5-specific CD8+ and CD4+ T cell responses, revealing a skewed Th1 phenotype and attenuated cytokine secretion.

The methodology, data interpretation and conclusions appear robust and valid. Even though there are recent reports on the XBB.1.5 variant, the current manuscript offers key findings and contributes to the understanding of its virological characteristics and its interactions with the immune system. Overall, the manuscript's clarity, context, and methodological rigor contribute to its potential significance. However, providing more explicit connections to recent relevant literature could enhance the manuscript's contextualization.

We thank the reviewer for their feedback and appreciate the comment regarding the manuscript's clarity, context, and methodological rigor. We have updated the manuscript to include explicit connections to recent literature that has been published since the original submission, as outlined below.

Major points:

1. The authors describe the cryo-EM structure of the XBB.1.5 spike ectodomain with an overall architecture that shows similarity with previous variants. Tamura et al. has recently determined the

structures of XBB.1 S ectodomain alone and the XBB.1 S-ACE2 complex by cryoelectron microscopy analysis. The reference of this paper is missing, and the authors should expand this section by providing a more detailed description on the structural features of XBB.1.5 spike protein. Alternatively, the authors could consider to combine the first two results by describing the structural and biochemical features of XBB.1.5 Spike with ACE2 receptor.

We have added the citation to Tamura et al. and have expanded our description of the structural features of the XBB.1.5 spike ectodomain as suggested.

2. Other reports have demonstrated enhanced receptor binding by the XBB.1.5 spike protein compared to XBB.1 variant, indicating a higher affinity of XBB.1.5 for the ACE2 receptor with KD values of 3,4nM and 19nM, respectively (Yue et al. 2023). Yeast surface display assay showed that the dissociation constant value of XBB.1.5 S receptor-binding domain from the human ACE2 receptor is significantly (4-3 times) lower than that of XBB.1 S receptor-binding domain (Uriu et al. 2023). In the current manuscript, the authors observed a strong binding affinity of XBB.1.5 RBD to hACE2 receptor with KD 0,75nM. How do the authors rationalize discrepancies in nM affinity through consistent methodology?

Differences in the overall technique, methodology and data fitting are 3 critical factors that can have a profound impact on the absolute value for KD obtained when performing binding experiments.

Yeast display is a fundamentally different technique from SPR and may involve presentation of different epitopes, and therefore should not be expected to yield identical absolute values for KD for many reasons including the following:

- (1) differences in RBD constructs (free soluble RBD vs RBD fusion constructs for display at the surface of yeast cells),
- (2) differences in data type, (SPR involves measuring the kinetics of the interaction and generates temporal data as proteins associate and dissociate, yeast display is an endpoint approximation of binding),
- (3) differences in the calculation of KD (SPR kinetic analysis involves estimating the ratio of K_{off}/K_{on} while yeast display approximates the KD as the EC50 of the endpoint saturation curve

Even when considering techniques such as SPR or BLI, there is considerable spread in the absolute numerical KD value obtained for binding interactions between spike protein constructs and ACE2. This can be due to several differences including:

- (1) The assay setup - which protein was immobilized, and what chemistry was used to immobilize the protein. Swapping ligand and analyte identities can change the calculated KD value, using covalent immobilization versus affinity-based immobilization can also have an impact on the calculated KD value.
- (2) The concentration series utilized for the analyte can have a large impact on the calculated KD value (ie an experiment using the same analyte at 500nM 250nM 125nM will generate a different KD value than at 300nM 100nM 33nM)
- (3) The duration of the association and dissociation phase can have a large impact on the calculated KD value (ie an experiment using the same analyte at the same concentrations using 60 seconds for association and 5 min for dissociation will yield different KD values than the same run using 180 seconds for association and 10 min for dissociation).
- (4) The protein constructs used (monomeric versus dimeric ACE2 and RBD versus trimeric spike ectodomain) will have a large impact on the calculated KD.
- (5) The model used to fit the data will have a profound impact on the calculated KD (1:1 versus heterogenous, bivalent, and other fits will all yield different KD values).

Presented in the table below is a brief survey of the SARS-CoV-2 spike - ACE2 binding literature to appreciate the heterogeneity in reported KD values for the WT RBD - ACE2 interaction.

KD	Technique	Citation
24.4nM	SPR	https://www.ncbi.nlm.nih.gov/pmc/articles/PMC9212699/
4.8nM	SPR	https://www.nature.com/articles/s41586-020-2180-5#Sec10
62.6nM/74.4nM	SPR	https://elifesciences.org/articles/70658
38.2nM	BLI	https://www.science.org/doi/10.1126/science.abn8863
17nM	SPR	https://www.sciencedirect.com/science/article/pii/S002228362100276X
52nM	SPR	https://www.nature.com/articles/s41401-021-00735-z
87.9nM	SPR	https://www.nature.com/articles/s41392-022-00914-2
8.3nM	SPR	https://elifesciences.org/articles/69091

Given that changes in the technique, methodology, and data fitting can yield such heterogeneous results, we do not take our estimated KD values as absolutes, rather we always perform comparisons to known constructs (ie WT RBD), and therefore comment on relative changes in binding affinity within the same experimental setup, using identical protocols and data fitting strategies.

3. The authors describe that the affinity of the XBB.1.5 spike for hACE2 is enhanced to a similar extent as BA.2, when compared to WT. With that regards, it would be preferable to include the XBB.1 RBD in the surface plasmon resonance (SPR) experiments. By doing so, the authors could gain a clearer and more comprehensive understanding of the described enhanced binding affinity for the XBB.1.5 spike proteins compared to the XBB.1 variant, as well as the contribution of the S486P mutation.

We agree with the reviewer and have performed the requested experiment, this time using BLI (in experimental triplicates), with the addition of the XBB.1 RBD. As shown below, we see a clear trend towards enhanced binding of ACE2 by XBB.1.5 compared to XBB.1. Given other reports further supporting this increase, we conclude that it is likely that the S48P mutation enhances binding affinity for ACE2. We have included this result in the revised manuscript (Supplemental Figure 2).

4. The authors well explain the structural basis of the receptor binding by cryo-EM studies of XBB.1.5 in complex with hACE2. Zhang et al. have recently conducted crystallographic studies on the receptor-binding domains (RBDs) from the subvariants XBB.1 and XBB.1.5, each complexed with human ACE2. Does the cryo-EM and crystal-based models superimpose well or there is a structural variation? This additional comparison would provide valuable insights into the interactions observed, i.e., the favorable hydrophobic interaction with M82 and L79. The study indeed underscores phenylalanine as the best adapted to hACE2 at position 486, followed by a proline and then a serine.

We have performed the suggested structural alignment and find that the published chimeric structures (SARS-CoV-2 receptor binding motifs grafted onto a SARS-CoV-1 RBD core) of XBB.1 and XBB.1.5 RBD – hACE2 solved by X-ray crystallography and our cryoEM structure reported in the present manuscript align well (Supplemental Figure 4 in the revised manuscript). Specifically, there is limited structural variation at the amino acid positions discussed in the manuscript (positions 493 and 486 within the RBD and 31, 34, 35, 79, and 82 within hACE2). We have added a sentence discussing this structural alignment within the results section in the revised manuscript.

5. The authors demonstrate enhanced retention of S309 binding to the XBB.1.5 spike protein - as compared to BA.2 - potentially due to the differential G339D (BA.2) versus G339H (XBB.1.5) mutation, which occurs within the S309 epitope. Cao et al. highlighted that, even though D339H is a charge-reversing mutation on the S309 binding interface, the neutralizing activity of S309 was not affected and even exhibited slightly improved neutralization activity against BA.2.75 compared with BA.2 and BA.4/5.

Recently, Qu and coworker also showed enhanced resistance of the sensitivity by XBB variants to neutralization by S309, highlighting the role of R346, G339H and L368I mutations occurred in XBB.1.5 spike variant. The authors should include this recent work in the discussion.

We thank the reviewer for drawing our attention to these reports which rationalize the different potency of S309 across Omicron and XBB variants. We have added discussion and reference to both of these papers to contextualize our findings within the revised results section.

Other comments:

1. The phrase "sought to" appears frequently in the manuscript; please replace it with more dynamic verbs.

We have replaced this phrase at multiple points throughout the manuscript.

2. The major findings outlined in the discussion, listed from 1 to 4, could be presented in more narrative fashion.

We appreciate this suggestion, however in this paragraph we are striving to very succinctly summarize the major findings of the manuscript to focus the majority discussion on contextualization of the results within broader themes of SARS-CoV-2 biology.

3. Receptor binding studies included in the manuscript demonstrate the retention of binding contacts with the human ACE2 receptor by XBB.1.5 variant. In the discussion, the authors summaries this result as increased XBB.1.5 spike - hACE2 affinity as compared to WT spike and similar affinity as compared to other Omicron variant spikes. To align with this conclusion the authors might consider rephrasing result's title "Enhanced Binding of Human ACE2 by the XBB.1.5 Spike Protein", thus enhancing the title coherence with the study's content.

We agree with the reviewer and have updated the manuscript to reflect this suggested change.

REVIEWERS' COMMENTS

Reviewer #1 (Remarks to the Author):

Thank you for a nice revision of the manuscript. From my perspective, you have satisfactorily responded to the comments and questions.

Reviewer #2 (Remarks to the Author):

The revised manuscript has significantly improved and has excellently addressed the comments of this reviewer.

Reviewer #3 (Remarks to the Author):

The authors have address all of the comments I raised during the initial review of the manuscript.

Reviewer #4 (Remarks to the Author):

The authors have addressed my previous concerns regarding the methodology and updated the discussion with state-of-the-art literature.

I appreciate the authors' detailed response to my comments and the efforts they have made to address the concerns raised during the initial review.

However, I would appreciate a further clarification on the major point n.3. The authors performed the requested experiment with the addition of XBB.1 RBD in triplicate. They described a clear trend towards enhanced binding of ACE2 by XBB.1.5 compared to XBB.1.

However, the observed differences did not achieve statistical significance, as noted in the result session. For this reason, I would suggest the following change:

“Recent reports have demonstrated enhanced receptor binding by the XBB.1.5 spike protein compared to its XBB.1 predecessor. We performed biolayer interferometry (BLI) experiments, finding an overall trend consistent with increased human ACE2 (hACE2) affinity for the XBB.1.5 RBD compared with XBB.1, although in our experimental setup this

difference did not attain statistical significance (Supplemental Figure 2A)".

Overall, the revisions have improved the clarity and the rigor of the manuscript.

Reviewer #1 (Remarks to the Author):

Thank you for a nice revision of the manuscript. From my perspective, you have satisfactorily responded to the comments and questions.

We thank the reviewer for their effort and assistance with this manuscript

Reviewer #2 (Remarks to the Author):

The revised manuscript has significantly improved and has excellently addressed the comments of this reviewer.

We thank the reviewer for their effort and assistance with this manuscript

Reviewer #3 (Remarks to the Author):

The authors have address all of the comments I raised during the initial review of the manuscript.

We thank the reviewer for their effort and assistance with this manuscript

Reviewer #4 (Remarks to the Author):

The authors have addressed my previous concerns regarding the methodology and updated the discussion with state-of-the-art literature. I appreciate the authors' detailed response to my comments and the efforts they have made to address the concerns raised during the initial review. However, I would appreciate a further clarification on the major point n.3. The authors performed the requested experiment with the addition of XBB.1 RBD in triplicate. They described a clear trend towards enhanced binding of ACE2 by XBB.1.5 compared to XBB.1. However, the observed differences did not

achieve statistical significance, as noted in the result session. For this reason, I would suggest the following change:

“Recent reports have demonstrated enhanced receptor binding by the XBB.1.5 spike protein compared to its XBB.1 predecessor. We performed biolayer interferometry (BLI) experiments, finding an overall trend consistent with increased human ACE2 (hACE2) affinity for the XBB.1.5 RBD compared with XBB.1, although in our experimental setup this difference did not attain statistical significance (Supplemental Figure 2A)”.

Overall, the revisions have improved the clarity and the rigor of the manuscript.

We have made the requested textual change to the manuscript. We thank the reviewer for their effort and assistance with this manuscript.